# Assembly mechanism of the inflammasome sensor AIM2 revealed by single molecule analysis

Meenakshi Sharma [1] & Eva de Alba [1] ✉

Pathogenic dsDNA prompts AIM2 assembly leading to the formation of the inflammasome, a multimeric complex that triggers the inflammatory response. The recognition of foreign dsDNA involves AIM2 self-assembly concomitant with dsDNA binding. However, we lack mechanistic and kinetic information on the formation and propagation of the assembly, which can shed light on innate immunity's time response and specificity. Combining optical traps and confocal fluorescence microscopy, we determine here the association and dissociation rates of the AIM2-DNA complex at the single molecule level. We identify distinct mechanisms for oligomer growth via the binding of incoming AIM2 molecules to adjacent dsDNA or direct interaction with bound AIM2 assemblies, resembling primary and secondary nucleation. Through these mechanisms, the size of AIM2 oligomers can increase fourfold in seconds. Finally, our data indicate that single AIM2 molecules do not diffuse/ scan along the DNA, suggesting that oligomerization depends on stochastic encounters with DNA and/or DNA-bound AIM2.

The innate immune system recognizes cues associated with cellular damage and molecular patterns arising from invading pathogens[1]. Recognition by sensor proteins triggers the assembly of large signaling platforms known as inflammasomes that activate inflammatory caspases[2–4]. Inflammasome formation involves sensor oligomerization[5–7], leading to the self-assembly of the adaptor protein ASC (Apoptosis-associated Speck-like protein containing a CARD) into the so-called "ASC speck"[8,9]. ASC recruits the effector procaspase 1[2,10], increasing its local concentration and promoting caspase activation[11], thus resulting in the maturation of proinflammatory cytokines[12] and cell death by pyroptosis and PANoptosis[13,14]. In the process of inflammasome activation, a filamentous punctum (ASC speck) with a diameter of ~ 0.5–1 µm forms via self-association and oligomerization of multiple protein components (Supplementary Fig. 1)[9]. At the molecular level, it has been shown that the inflammasome adaptor ASC and its isoform ASCb[8] with two oligomerization Death Domains, PYD (Pyrin Domain) and CARD (Caspase Activation and Recruitment Domain), can polymerize into different macrostructures[15–18]. ASC connects PYD-

containing sensors[19] and procaspase 1 via homotypic interactions to facilitate speck assembly and activation[20].

Inflammasome sensors show specificity for different molecular patterns. For instance, foreign dsDNA activates the sensors AIM2 (Absent In Melanoma 2) and IFI16 (Interferon gamma Inducible protein 16)[21–24]. Both sensors carry an N-terminal PYD for self-assembly and polymerization with ASC (Supplementary Fig. 1) leading to the formation of the inflammasome, and a C-terminal HIN (Hematopoietic, Interferon-inducible, Nuclear localization) domain(s) for DNA binding. However, AIM2 is a cytosolic sensor, whereas IFI16 is the only sensor identified thus far that recognizes foreign DNA in the nucleus[25,26]. Detailed functional and structural studies of complexes between dsDNA and the DNA binding domains of the cytosolic and nuclear sensors explain the lack of sequence specificity, as the intermolecular interactions involve the dsDNA phosphate backbone[25]. Importantly, the authors indicate that dsDNA serves as an oligomerization platform for the inflammasome and inform on the estimated size of the oligomers. Specifically, the X-ray structure shows 2 HIN domains of AIM2 bound to a 20-mer dsDNA[25] (Supplementary Fig. 1).

[1]Department of Bioengineering, School of Engineering, University of California Merced, Merced, California, USA.
✉e-mail: edealbabastarrechea@ucmerced.edu

Equally elegant studies on the function and operating modes of AIM2 and IFI16 found that DNA binding and sensor self-association are integrated and cooperative processes[26,27]. The PYD domain was found to be essential for these functions and specifically required for strong binding to dsDNA and polymerization in the presence of excess dsDNA. These studies show that AIM2-DNA and IFI16-DNA binding affinity depends on the DNA length, as the affinity increases steeply for dsDNA longer than a threshold of ~70 bp (hosting ~6 AIM2 protomers) until reaching a maximum value for ~280 bp DNA (hosting ~24 AIM2 protomers)[27]. This work thus indicates that the DNA acts as a molecular ruler for AIM2 inflammasome assembly following a switch-like mechanism[27].

Furthermore, the interaction between the nuclear sensor IFI16 and dsDNA has been studied using single molecule fluorescence imaging by TIRF microscopy (Total Internal Reflection Fluorescence)[28]. This study shows single IFI16 molecules diffusing several µm along the λ-phage dsDNA. IFI16 scans the dsDNA to find other molecules already bound to DNA for oligomerization. A sufficiently long stretch of free dsDNA is required for scanning and oligomerization, thus elegantly explaining how IFI16 discriminates between self- and foreign-DNA, as the former does not expose sufficiently long dsDNA fragments available for self-assembly due to nucleosome packing[28].

Despite the challenges associated with protein oligomerization, combined efforts using a variety of biophysical, biochemical, and microscopy techniques are significantly advancing our understanding of AIM2 inflammasome formation[25,27–30]. However, there are unresolved questions on the mechanistic and kinetic aspects of the AIM2-DNA assembly process. First, we do not know whether AIM2 forms individual, distinct oligomers on the DNA, and in this case, whether oligomers of different sizes can coexist and what their shapes are. Second, what are the association and dissociation rates of a single AIM2-DNA complex? Third, how fast do oligomers grow and what mechanisms are followed for oligomerization and propagation? Finally, since the function of AIM2 does not require to discriminate between self and foreign DNA, does the cytosolic sensor follow the same scanning mechanism as the nuclear sensor IFI16? Here, we provide answers to these questions by combining optical traps and confocal fluorescence microscopy to study the initial stages of the AIM2 inflammasome.

## Results

### AIM2 forms oligomers of different sizes and shapes bound to dsDNA

The typical experimental setup using optical traps and confocal fluorescence microscopy is shown in Fig. 1a. A single λ-phage dsDNA molecule (~16.5 µm long) is tethered between two optically trapped beads (∅ ~ 3 µm) and moved to the protein channel of the microfluidics flow cell. Two-dimensional (2D) images are acquired by laser scanning confocal fluorescence microscopy. In seconds, AIM2 labeled with the fluorophore Alexa 488 forms clusters of various sizes that populate

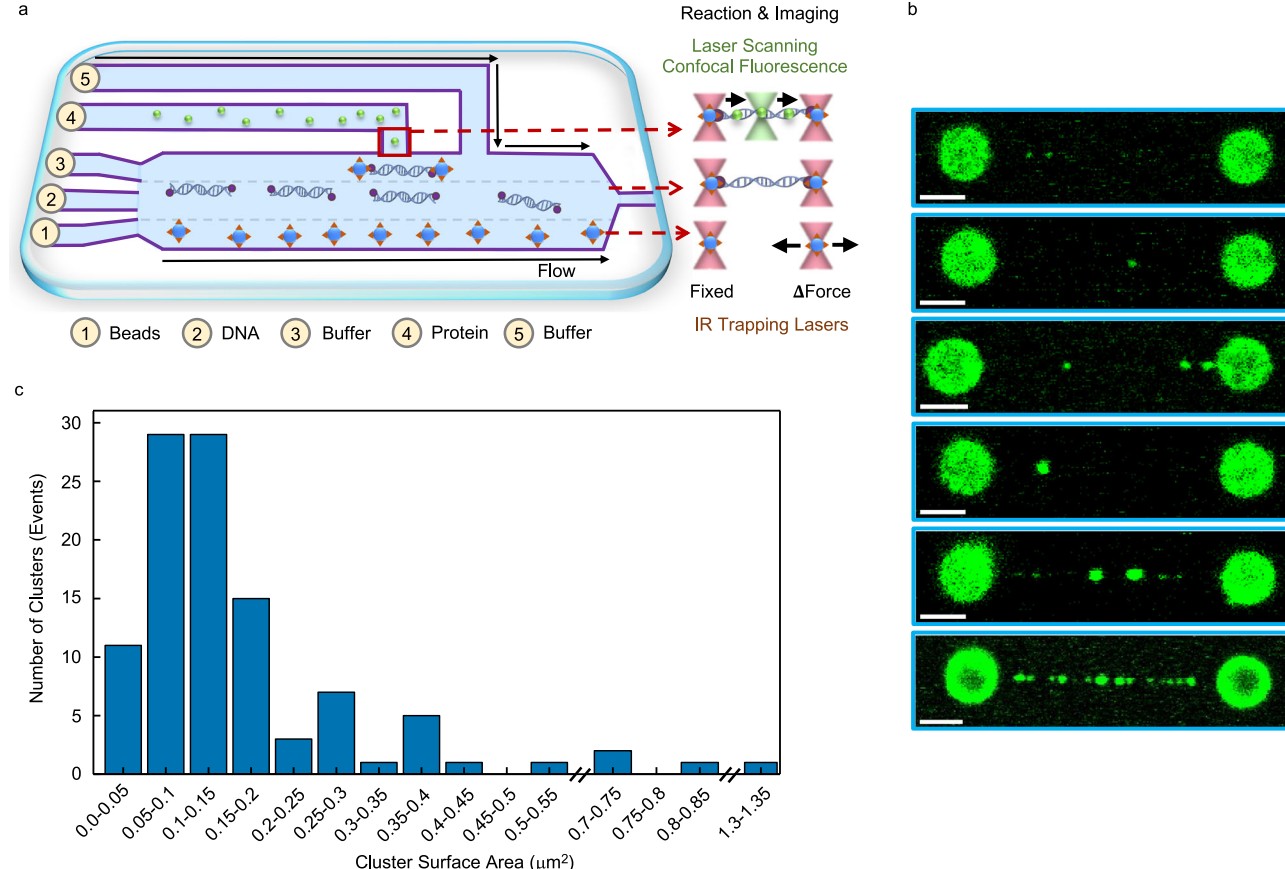

**Fig. 1 | AIM2 forms distinct oligomers of different sizes and shapes bound to a single dsDNA molecule. a** Schematic representation of a typical experimental setup, showing the microfluidics flow cell with streptavidin-coated beads in channel 1, biotinylated λ-phage dsDNA in channel 2, buffer in channel 3, and fluorescent AIM2 in channel 4. Channels 1–3 are subjected to laminar flow. The beads trapped with infra-red (IR) trapping lasers (red cones) in channel 1 are moved to channel 2 to tether the dsDNA molecule. Subsequently, the beads are moved to channel 3 to perform force-extension measurements to determine the number of dsDNA molecules attached. Imaging is done by confocal fluorescence microscopy in channel 4 (green cones). Thin and thick arrows show the direction of flow and laser movement, respectively. **b** Representative examples of multiple (*n* > 278) two-dimensional fluorescence scans obtained seconds after exposing the dsDNA to the protein channel. All scale bars represent 3 µm. **c** Cluster size distribution based on surface area (*n* = 106). Source data are provided as a Source Data file.

multiple positions of the dsDNA molecule at sub-nanomolar to low nanomolar protein concentrations (~0.2–14 nM) under physiological salt concentration (160 mM KCl) (Fig. 1b). In some instances, small AIM2 clusters span the DNA molecule and in other cases, only a few large clusters appear. Large and small clusters are observed together (Fig. 1b). The multiple binding events along the dsDNA molecule suggest that AIM2 lacks sequence specificity, as demonstrated by the X-ray structural studies on the AIM2[HIN]-dsDNA complex[25].

An analysis of the surface area distribution of single AIM2 particles reveals that cluster sizes are typically smaller than 0.2 μm² (Fig. 1c) with most clusters falling in the 0.05–0.15 μm² range. The cluster surface area does not represent the actual size of the oligomer due to the optical resolution of the confocal microscope and the filamentous nature of AIM2 polymers[27]. For example, a 250-mer dsDNA will host ~25 AIM2 molecules based on the X-ray structure of the complex[25] and span 85 nm (0.34 nm per base pair). However, a complex of this size will result in a fluorescent spot of larger dimensions.

To estimate the expected dimensions of a dsDNA fragment of 85 nm, Supplementary Fig. 2a shows the fluorescent image and Point Spread Function (PSF) of a commercial fluorescent bead with a diameter of 100 nm. The FWHM (Full Width at Half Maximum) of the PSF associated with this image is 351 and 312 nm in the X- and Y-axis, respectively. Analogously, a smaller bead (23 nm) results in a fluorescent spot with FWHM (X, Y) of 342 and 322 nm, respectively (Supplementary Fig. 2b). Clusters in the 0.05–0.15 μm² range correspond to diameters of ~250–440 nm, which are close to the FWHM value of the 100 nm bead. Thus, these dimensions indicate that most cluster sizes could host approximately 25 molecules or less. However, the filamentous AIM2 oligomers likely lead to the different cluster shapes observed and could result in surface areas that do not properly represent the number of protomers in the cluster.

Overall, this analysis shows that AIM2 oligomers of different sizes and shapes coexist bound to dsDNA. Our results suggest that AIM2 oligomerization upon dsDNA binding can occur in a variety of macrostructural arrangements that might influence the overall assembly of the inflammasome.

## AIM2 oligomers bound to dsDNA are typically smaller than 25 molecules

We have estimated the number of molecules in the different clusters using fluorescence intensity relative to the intensity produced by a single fluorophore. Several assumptions were made to correlate fluorescence intensity with the number of fluorophores. Specifically, we assume that the detector response is close to linear due to the low dead time (35 ns) of the Avalanche Photodiode Detector (APD) (Methods). This dead time results in ~8% underestimation of photon counts for a cluster of 10 emitting fluorophores assuming an idealized dead time model in which the detector is not affected by events happening during the dead time[31] (calculations of the underestimation percentage are described in Methods). Photon count underestimation leads to an error of ~1 fluorophore in a 10-fluorophore cluster. An approximately linear APD response is achieved by working under conditions that avoid detector saturation (i.e., low laser power (10%) and low number of photons detected due to the confocal setup). In addition, both the numerical aperture of the objective and the confocal microscopy setup restrict the angles at which photons are detected. This effect may be ignored for fluorophores with isotropic rotation. However, isotropic motion might be compromised in the presence of AIM2 oligomerization, thus photons emitted by fluorophores positioned at the appropriate angles might have greater chances of being detected. Our estimations do not consider this effect. Additionally, we can safely assume that laser excitation (at constant power) is uniform across the ROI (Region of Interest) by scanning the confocal plane. With respect to the optical axis, we assume the detector collects photons emitted by assembled molecules lying

within the Z-axis resolution of the microscope (~1 μm). This assumption might result in a slight underestimation of the number of molecules due to variations in excitation along the Z-axis even if clusters are rarely larger than 1 μm in diameter.

In addition, correction factors need to be applied to consider intensity decay due to photobleaching and fluorophore labeling efficiency. Evidence show that fluorophore clustering affects the emission properties of individual fluorophores due to energy transfer processes between them[32–34]. In fact, homo-FRET[35] and Stokes shift processes[36] resulting from fluorophore clustering have been observed. Therefore, the total emitted light of fluorophores clustered in proximity is frequently enhanced or decreased relative to the expected intensity from the total number of fluorophores[32, 37]. For these reasons, fluorophore clustering poses challenges in determining the stoichiometry of protein complexes based on fluorescence intensity[37–40]. AIM2 labeled with Alexa 488 is expected to emit at 520 nm (microscope emission filter at 512 nm and bandwidth of 25 nm); however, to analyze the effect of AIM2 oligomerization on fluorescence emission, we have also detected emission at red wavelengths (microscope emission filter at 700 nm and bandwidth of 100 nm). Surprisingly, we have observed emission in red for AIM2 clusters emitting more than 100 blue photon counts (Supplementary Fig. 3a). The number of red photon counts remains relatively constant with an average of 14 ± 4 photons for clusters emitting up to 575 blue photons and typically increases non-linearly for larger oligomers (Supplementary Fig. 3b). This behavior suggests that the emission at red wavelengths is not bleed through.

We have observed that the fluorescence intensity emitted by AIM2 oligomers bound to dsDNA decreases with time due to photobleaching (Supplementary Figs. 4 and 5). Decay rates of the fluorescence intensity produced by AIM2 clusters vary from 0.2 s⁻¹ to 0.5 s⁻¹ with an average of 0.43 s⁻¹ for clusters emitting less than 1000 photon counts. Based on these decay rates, the effect of photodepletion is negligible during the short excitation time of the scanning laser in 2D scans (~5 ms for an ROI of 1 μm²).

The fluorescence intensity was corrected considering the fluorophore labeling efficiency, which was determined to be approximately 90% using mass spectrometry (Methods section). The initial fluorescence intensity divided by the intensity emitted by a single fluorophore allows us to estimate the number of molecules per oligomer. The photon counts emitted per fluorophore were determined in two ways: (1) by the identification of photobleaching steps in fluorescence intensity decays (Methods, Supplementary Figs. 4, 5)[41,42]; (2) by detecting the photon counts emitted by the single fluorophore, Atto 488, attached to the dsDNA (Supplementary Fig. 6). The molecular structures of Alexa 488, used to label AIM2, and Atto 488, are very similar (Supplementary Fig. 6d) as well as their quantum yields[43–46]. The resulting histogram distributions using these two methods show that 11 ± 4 and 12 ± 2 photon counts, respectively, are more frequent, thus corresponding to a single fluorophore (Supplementary Figs. 4b, 6c). As expected, the intensity of single fluorophores is steady with time (Supplementary Fig. 4c, d). In addition, to consider the loss in blue photons in the 100–575 range due to fluorophore emission in red, we have assumed that the average of 14 red photons is equivalent to one fluorophore.

The distribution of the number of molecules per oligomer resulting from the analysis of 155 clusters up to ~100 molecules shows that oligomers smaller than 25 molecules are more abundant (Fig. 2a). There is no clear predominance of a specific oligomer size within this range, except for clusters composed of 13–15 molecules being slightly more frequent (Fig. 2b). Within associated errors, these results are in accord with previous studies indicating a preferred oligomer size of 20–24 AIM2 molecules[25,27]. The number of molecules in clusters emitting more than 575 photons is only an approximate value as energy transfer processes are happening more pronouncedly than for smaller clusters (Supplementary Fig. 3). The direct visualization of individual oligomers reveals the coexistence of small and large AIM2

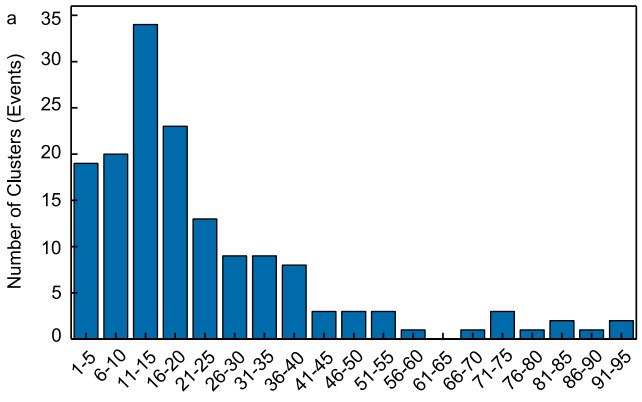

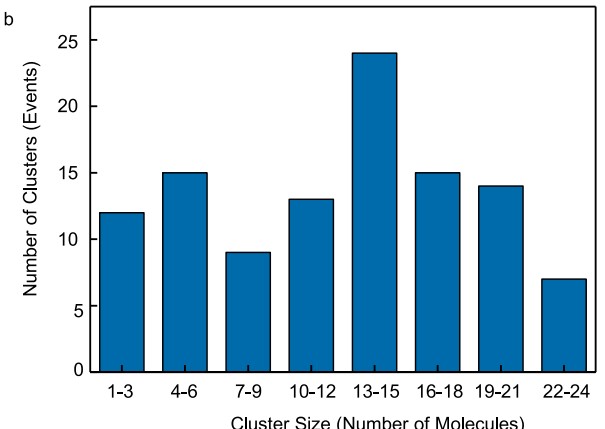

**Fig. 2 | AIM2 oligomers attached to a single dsDNA are predominantly smaller than 25 molecules. a** Cluster size distribution based on the number of molecules up to ~100 molecules and **b** up to 25 molecules ($n = 155$). Oligomers of 13–15 molecules are slightly more frequent. The number of molecules associated with clusters emitting photon counts in the 575–1500 range (~52–136 molecules) is likely underestimated due to the increased emission at red wavelengths (Supplementary Fig. 3). Values of oligomer size carry propagated error from the photon count analysis. The smallest error of 17% in these measurements (Supplementary Fig. 6) leads to an error in cluster size; $\triangle S = 0.17 S$, where $S$ is the size of the cluster (i.e., the number of molecules). Source data are provided as a Source Data file.

clusters bound to dsDNA (Figs. 1 and 2), thus providing additional insight into an all-or-none process previously proposed for AIM2-DNA binding[27].

Overall, clusters larger than 50 molecules are less frequent (Fig. 2a). In addition, some large clusters have been observed both bound to dsDNA and trapped together with the beads when there is no DNA (Supplementary Fig. 7). We believe AIM2 might undergo slight oligomerization in the absence of dsDNA during the single molecule experiments, even though AIM2 elutes as a monomer at 10–14 nM during the last protein purification step (Supplementary Fig. 8, Methods). It has been reported using ns-TEM (Transmission Electron Microscopy) that AIM2 polymerizes into filaments in the absence of DNA at concentrations greater than 500 nM[27]. After extensive ns-TEM analysis of AIM2 solutions at 10 nM, we were not able to clearly observe filaments likely due to the low concentration. This result suggests that the oligomeric species are a minor population; however, the use of single molecule techniques might have facilitated the detection of these assemblies.

**AIM2-DNA dissociation rate constant at the single molecule level**
Experiments using confocal fluorescence microscopy and optical tweezers allow to analyze the association and dissociation kinetics

between AIM2 and the single dsDNA molecule (Fig. 3a). Specifically, we determined the time single AIM2 molecules and AIM2 self-assemblies remain bound to the dsDNA molecule. For this purpose, the fluorescence intensity (photon counts) of fluorophore tagged AIM2 is recorded as a function of time and position on DNA in kymographs. For these experiments, the λ-dsDNA molecule is mechanically controlled by the optical traps (Fig. 1a) and stretched at a constant length of 16 μm using a force of 17 pN. Figure 3b shows a representative sample of typical kymographs obtained for AIM2 oligomers of different sizes with traces of different intensity and retention times. To differentiate between AIM2 oligomerization and DNA binding, we determined the residence times of hundreds of single molecule traces ($n = 314$) obtained at 1 nM and 5 nM protein concentrations, including few traces at 2 nM and 10 nM (Fig. 3c, d). These traces correspond to an average of 10 photon counts, thus in good agreement with the 11 photon counts determined by the photobleaching step analysis (Methods, Supplementary Figs. 4, 5) and by the single fluorophore attached to λ-DNA (Methods, Supplementary Fig. 6). The residence time distribution of the single molecule traces was used to determine the AIM2-DNA dissociation rate constant ($k_{off}$) by fitting the histogram to an exponential equation (Fig. 3d). The $k_{off}$ for single AIM2 molecules to dissociate from the dsDNA molecule is $0.29 \pm 0.01 \, s^{-1}$. To our knowledge, this is the first dissociation rate constant obtained for a dsDNA inflammasome sensor. This value is approximately two orders of magnitude larger than typical values reported for transcription factors[47,48]. A higher tendency of inflammasome DNA sensors to detach from DNA could be expected based on the known lack of sequence specificity[25]. Interestingly, similar $k_{off}$ values have been observed for the Klenow fragment of *E. coli* polymerase I ($0.40 \pm 0.01 \, s^{-1}$), which does not require sequence specificity for binding to the template DNA[49].

To determine the potential effect of protein concentration on the dissociation rate, we have selected approximately 100 single molecule traces at both 1 nM and 5 nM from the previous data (Fig. 3e, f). The $k_{off}$ values obtained at the two concentrations are $0.29 \pm 0.02 \, s^{-1}$ and $0.33 \pm 0.02 \, s^{-1}$, respectively. These values are similar to the $k_{off}$ obtained with the traces acquired at different concentrations ($0.29 \pm 0.01 \, s^{-1}$), thus indicating that the dissociation rate of the AIM2-DNA complex does not depend on the protein concentration.

We have observed that AIM2 clusters composed of 3 or more protomers are permanent and thus remain attached to the dsDNA molecule for the total length of the kymographs, longer than 20 minutes in some instances (Supplementary Table 1 and Fig. 3b). These results indicate that self-assembly is critical to modulating the $k_{off}$ of AIM2-dsDNA complexes. Altogether, the dissociation data of AIM2-DNA support the concept of inflammasome formation being almost an irreversible process[29] for sufficiently large oligomers.

To investigate whether dsDNA stretching affects AIM2 binding, we have increased the force pulling the trapped beads to 40 pN leading to an end-to-end distance of the λ-DNA of 16.5 μm. We analyzed single AIM2 molecule traces ($n = 172$) in kymographs acquired under these conditions. The $k_{off}$ obtained after dwell time analysis is $0.52 \pm 0.08 \, s^{-1}$ (Supplementary Fig. 9). This result indicates that the additional force reduces the residence time of AIM2 bound to dsDNA, raising the question of whether potential distortions of the dsDNA structure affect AIM2 binding. In fact, stretching the dsDNA with small forces (20 pN) leads to a decrease in the inter-strand distance, resulting in dsDNA overwinding as shown by studies using all-atom molecular dynamics simulations[50]. In addition, several studies have been reported on the effect of dsDNA stretching on protein function. For example, it has been shown that dsDNA stretching at forces ranging from 5 to 40 pN induces off-target activity of the endonuclease Cas9 due to structural disruptions in the dsDNA molecule[51]. An additional example is RAD51 recombinase which forms nucleoprotein filaments on ssDNA and dsDNA displaying higher nucleation rates on dsDNA at forces increasing from 20 to 50 pN[52].

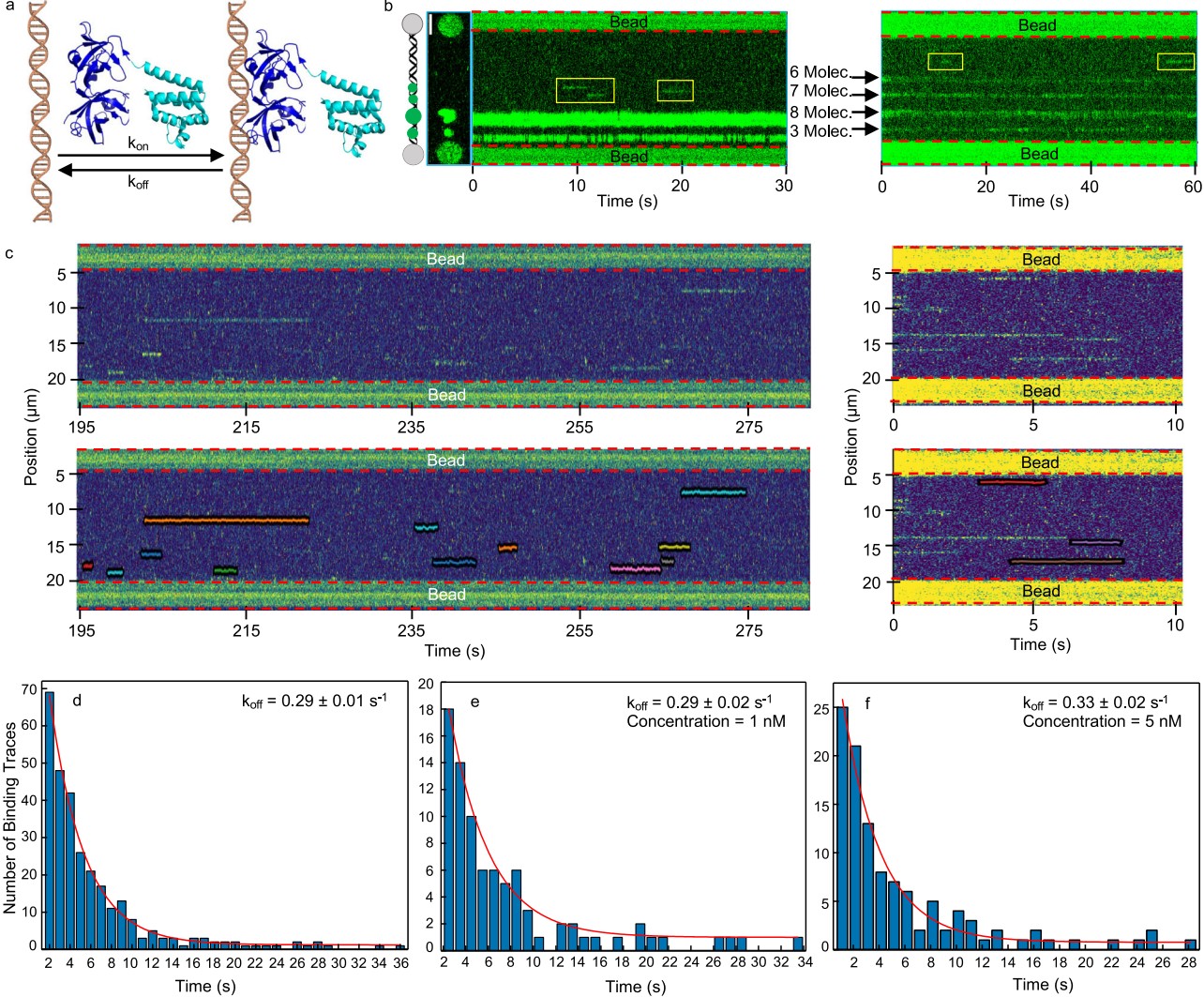

**Fig. 3 | Kinetics of the association and dissociation of single AIM2 molecules to dsDNA. a** Schematic representation of the association and dissociation of AIM2 to dsDNA (shown at different scales) and the corresponding rate constants ($k_{on}$ and $k_{off}$). The structures of the PYD[61, 62] and HIN[25] domains of AIM2 are shown in cyan and dark blue, respectively. **b** Examples of two kymographs ($n = 47$) showing the coexistence of traces corresponding to large clusters and single molecule traces (kymograph at the left with the corresponding 2D image), and small clusters composed of the indicated numbers of protomers (kymograph at the right). Single molecule traces are encompassed by yellow boxes. Scale bar represents 3 μm. **c** Examples of kymographs (top) and the resulting single molecule trace tracking (bottom). Only traces appearing after the kymograph started are selected to avoid biases in determining the residence time on dsDNA (kymograph at the right). **d–f** Dwell time analysis of AIM2 attached to dsDNA from traces acquired mainly at 1 nM and 5 nM protein concentration, including few traces at 2 nM and 10 nM (**d**), only traces acquired at 1 nM (**e**) and 5 nM (**f**) protein concentration. The red lines represent the fittings to a single exponential function reporting the dissociation rate constant ($k_{off}$). The goodness of fit is represented by R-square and RMSE (root mean squared error) values of 0.99, 0.96, 0.97, and 1.5, 1.0, 1.1 for the three fittings at all concentrations, 1 nM and 5 nM, respectively. Molecule (Molec.). Source data are provided as a Source Data file.

## AIM2-DNA association rate constant at the single molecule level

Real-time fluorescence anisotropy measurements in bulk have been used to determine binding rates of full-length AIM2 and dsDNA[29]. These thorough studies report that full-length AIM2 assembles into 600 bp dsDNA with an observed rate of ~ 0.8 min$^{-1}$ (at 72 nM AIM2)[29]. The observed rate in bulk includes both the association rate ($k_{on}$) and $k_{off}$. To estimate a value of the $k_{on}$, we have assumed that the $k_{off}$ is zero as the AIM2 oligomers will be composed of approximately ~20 molecules[27] leading to permanent attachment based on our data (Fig. 3b and Supplementary Table 1). Under this assumption, the $k_{on}$ is 0.18·10$^6$ M$^{-1}$ s$^{-1}$ at the reported AIM2 concentration (Table 1).

To determine $k_{on}$ of full-length AIM2 on the single λ-dsDNA, we analyzed over 100 single molecule traces in kymographs acquired with a constant time length of 600 s. Importantly, the observed traces do not show attachment and detachment in the same position in the dsDNA, which is expected based on the absence of

sequence specificity. Therefore, it has not been possible to identify "unbound" periods ($t_{on}$) between traces of protein attached to the same position in the DNA, thus making this measurement challenging. Therefore, we have considered the "unbound" period as the total observation time minus the sum of the residence time of all traces for each individual kymograph (Supplementary Tables 2, 3). This analysis has been done at 1 nM and 5 nM starting protein concentration values.

We noticed kymographs acquired at 1 nM with more than 20 single molecule traces, whereas most kymographs acquired at 5 nM show 11–14 traces (Supplementary Tables 2, 3). This observation suggests that increasing the AIM2 concentration decreases the number of single molecules due to the slight oligomerization in the absence of dsDNA, thus influencing the $t_{on}$ values. Therefore, it is more reliable to report a range of association rate constant values ($k_{on}$) obtained at the two concentrations (Table 1). This interval agrees overall with the $k_{on}$

**Table 1 | Association and dissociation rates and equilibrium dissociation constant (K$_D$) of AIM2 and IFI16 variants to dsDNA**

| | k$_{off}$ (s$^{-1}$) | k$_{on}$ (M$^{-1}$ s$^{-1}$) | K$_D$ (nM) |
|---|---|---|---|
| Single molecule: AIM2$^{FL}$ | 0.29 ± 0.01 | 0.37·10$^6$–1.81·10$^6$ | 160–780 |
| Fluorescence Anisotropy: AIM2$^{FL}$ | N/A | 0.18·10$^{6a}$ | ≤ 3 |
| Fluorescence Anisotropy: AIM2$^{HIN}$ | N/A | N/A | 176 ± 35[b] |
| Fluorescence Anisotropy: AIM2$^{HIN}$ | N/A | N/A | 212 ± 28[c] |
| Fluorescence Anisotropy: MBP-AIM2$^{FL}$ | N/A | N/A | 234 ± 42[c] |
| Fluorescence Anisotropy: MBP-AIM2$^{HIN}$ | N/A | N/A | 584 ± 22[c] |
| Fluorescence Anisotropy: IFI16$^{FL}$ | N/A | N/A | 65 ± 19[d] |

[a]k$_{on}$ extracted from reported k$_{obs}$ ~ 0.8 min$^{-1}$ using FAM-dsDNA600, 72 nM AIM2$^{FL}$ and 160 mM KCl[29].
[b]FAM-dsDNA20 (200 mM NaCl)[25].
[c]FAM-dsDNA72 (160 mM KCl)[27].
[d]FAM-dsDNA72 (160 mM KCl)[26].

derived from the assembly rate determined by the fluorescence anisotropy experiments in bulk[29] (Table 1).

## AIM2-DNA equilibrium dissociation constant free of oligomerization effects

Using fluorescence anisotropy in bulk, the dissociation constant (K$_D$) between full-length AIM2 and DNA has been estimated to be ≤ 3 nM[27]. However, the K$_D$ increases to 212 ± 28 nM for the truncated construct lacking the PYD (AIM2$^{HIN}$)[27]. A similar K$_D$ value (K$_D$ = 176 ± 35 nM) has been reported for AIM2$^{HIN}$ [25] (Table 1). The K$_D$ values obtained in the presence and absence of the PYD suggest that higher affinity requires protein oligomerization via PYD[27]. However, truncation of the protein could result in unwanted structural and functional modifications. Thus, separating protein-DNA binding from protein oligomerization using the native sequence will report on the affinity of AIM2 for DNA devoid of effects from protein oligomerization.

The estimated k$_{on}$ values and the determined k$_{off}$ using single molecule analysis allow obtaining an approximate value of the affinity of AIM2 for dsDNA in the absence of self-association. Our single molecule results indicate that the K$_D$ of the AIM2-dsDNA complex falls in the sub-micromolar range (Table 1). These K$_D$ values are more than three orders of magnitude larger than the K$_D$ reported based on the fluorescence anisotropy studies for full-length AIM2[27], and close to the values reported for the truncated AIM2 lacking the PYD domain (Table 1)[25,27]. A possible explanation of this result stems from selecting binding events of single AIM2 molecules to dsDNA, thus not including PYD-mediated AIM2-AIM2 oligomerization.

## The size of DNA-bound AIM2 oligomers increases fourfold in seconds

Using real-time FRET (Förster Resonance Energy Transfer) and fluorescence anisotropy in bulk, it has been reported that assembly rates of full-length AIM2 and DNA depend on the length of the latter[27,29]. An increase in assembly rates close to 700-fold has been observed when extending dsDNA from 24 to 600 bp[29]. However, we lack information on oligomer size and oligomer growth rates in the presence of sufficiently long DNA.

To obtain this information, we have monitored the growth rate of AIM2 clusters bound to λ-phage dsDNA (48.5 kbp). The change in photon counts as a function of time reports on the growth of oligomers with different starting numbers of molecules: 1, 2, and 4 molecules (Fig. 4a–c). In some instances, the oligomers can quadruplicate their size in approximately 4 s (Fig. 4a, b). The lack of a steady increase in photon counts is likely due to fluorophore blinking; however, a clear upward trend is observed.

Oligomer growth is also detected by careful analysis of the change in photon counts as a function of time in kymographs of single AIM2 molecules and oligomers bound to dsDNA (Fig. 5). The large oligomer triplicates its size in 0.85 s growing from 11 to 35 molecules. The

decrease in intensity for oligomers larger than 3 molecules (Fig. 5) reflects photobleaching and does not indicate AIM2-DNA dissociation, as we have shown that oligomers of this size or larger are permanently bound. However, 2-molecule clusters can detach from the dsDNA (Fig. 5).

Oligomer growth rates for several representative clusters obtained as the difference between the final and initial photon counts divided by the total observation time tend to increase with AIM2 concentration: 7.4 s$^{-1}$, 10.7 s$^{-1}$, and 34 s$^{-1}$ at 0.5 nM, 2 nM, and 13.5 nM, respectively. However, the lack of linearity between growth rate and concentration indicates that other factors could play a role in the oligomerization rate. For example, the AIM2 monomer shown in the middle kymograph of Fig. 5 grows at a rate of 5.8 s$^{-1}$, whereas the dimer at the bottom kymograph forms a hexamer at a rate of 94 s$^{-1}$. Both kymographs were acquired at 1 nM AIM2. These results suggest that the starting number of molecules likely influences the oligomerization rate. In addition, the structural arrangements of the oligomers (Fig. 1b) will dictate the different interacting possibilities, which could also affect the growth rate. Additional representative examples of cluster growth are shown in Supplementary Fig. 10, with corresponding growth rates ranging from 11.4 s$^{-1}$ (cluster size increasing from 4 to 6 molecules at 2 nM) to 25.3 s$^{-1}$ (cluster size increasing from 4 to 11 molecules at 10 nM).

## DNA-bound oligomers grow via distinct mechanisms

Two-dimensional frames of confocal images captured in continuous scanning mode were analyzed to identify specific cluster growth directions. Average pixel brightness (gray values) calculated along the vertical axis of the 2D frame are represented as a function of distance along the DNA molecule resulting in intensity profile plots. The comparison of profile plots from 2D frames acquired at different times shows that AIM2 oligomers grow along both right and left directions of the dsDNA (Fig. 6a, b). This observation implies that oligomer growth happens by incoming AIM2 molecules binding to the dsDNA at either side of a particular cluster or previously bound molecule. This is an expected result considering that AIM2 does not show DNA sequence specificity and suggests that AIM2 binding to either side of a pre-existing oligomer is stochastic. Interestingly, we have observed an increase in fluorescence intensity not localized at either side of the dsDNA but in perpendicular direction (Fig. 6c). This data indicate that oligomer growth can happen by the interaction of incoming AIM2 molecules with DNA-bound oligomers instead of binding directly to the dsDNA (Fig. 6c). Since the function of the PYD in AIM2 is to participate in protein-protein interactions, this type of oligomer growth is likely driven by PYD-PYD binding and does not depend on interactions with the DNA. This type of oligomer would expose AIM2 molecules with free HIN domains available to interact with other dsDNA molecules or fragments. The possibility of AIM2 oligomers growing by

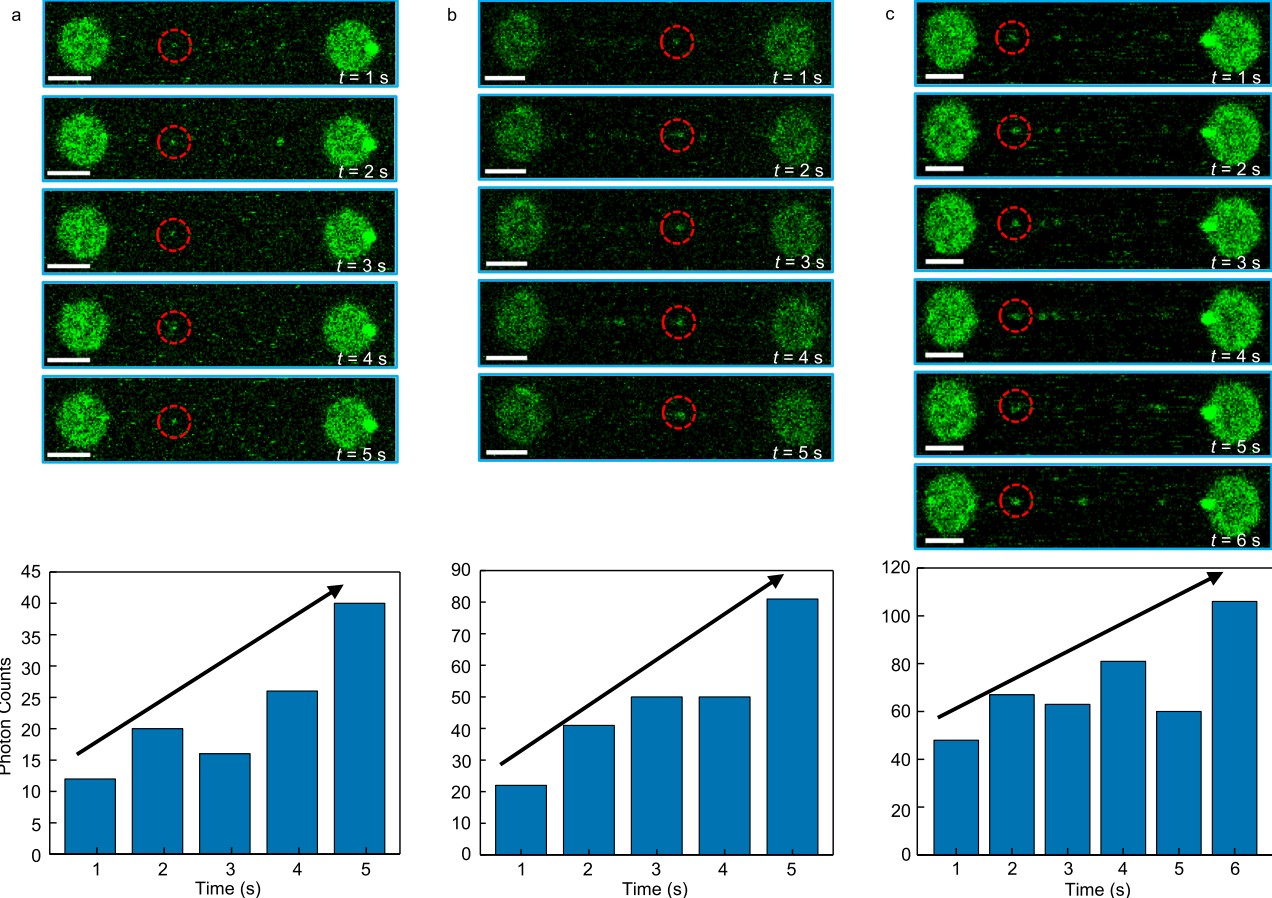

**Fig. 4 | Growth rate of AIM2 oligomers bound to dsDNA.** Top: Two-dimensional confocal scans (frames) of movies acquired at the times indicated (white) showing AIM2 binding to the single dsDNA. Starting number of molecules in the fluorescent spot circled in red are **a** 1; **b** 2 and **c** 4. Scale bar in white represents 3 μm. Data from (**a**–**c**) are representative of multiple movies (*n* = 45). Bottom: Overall increase in photon counts with time for the three fluorescent spots corresponding to 1, 2 and 4 starting molecules. AIM2 concentration is 5 nM (**a**, **b**) and 2 nM (**c**). Arrows represent an upward trend in the fluorescence intensity. Source data are provided as a Source Data file.

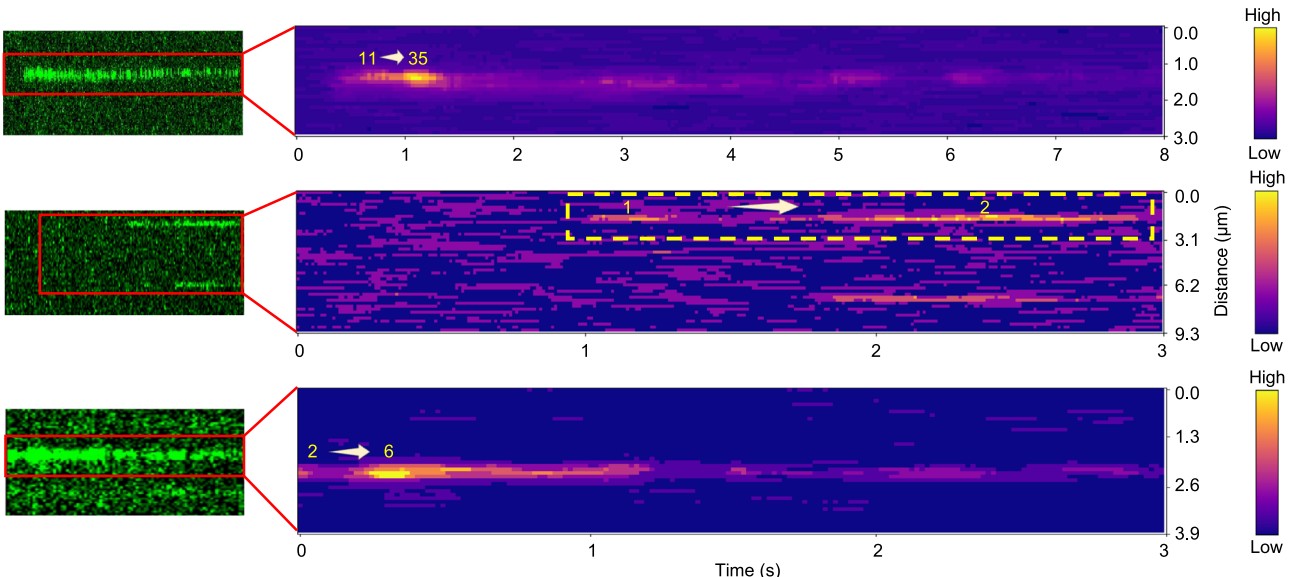

**Fig. 5 | Kymographs showing AIM2 oligomer growth.** Original kymographs of single AIM2 molecules and oligomers bound to tethered dsDNA (left) and their corresponding heat maps (right) showing an increase in fluorescence intensity as a function of time in some regions. The overall decrease in fluorescence intensity in most regions of the kymographs is due to photobleaching. The number of molecules corresponding to the photon counts is indicated in yellow. The heat map color bar is shown for clarity (right). AIM2 concentration: top (10 nM); middle and bottom (1 nM). The calculated growth rates are 302.5 s$^{-1}$ (top), 5.8 s$^{-1}$ (middle), and 94 s$^{-1}$ (bottom).

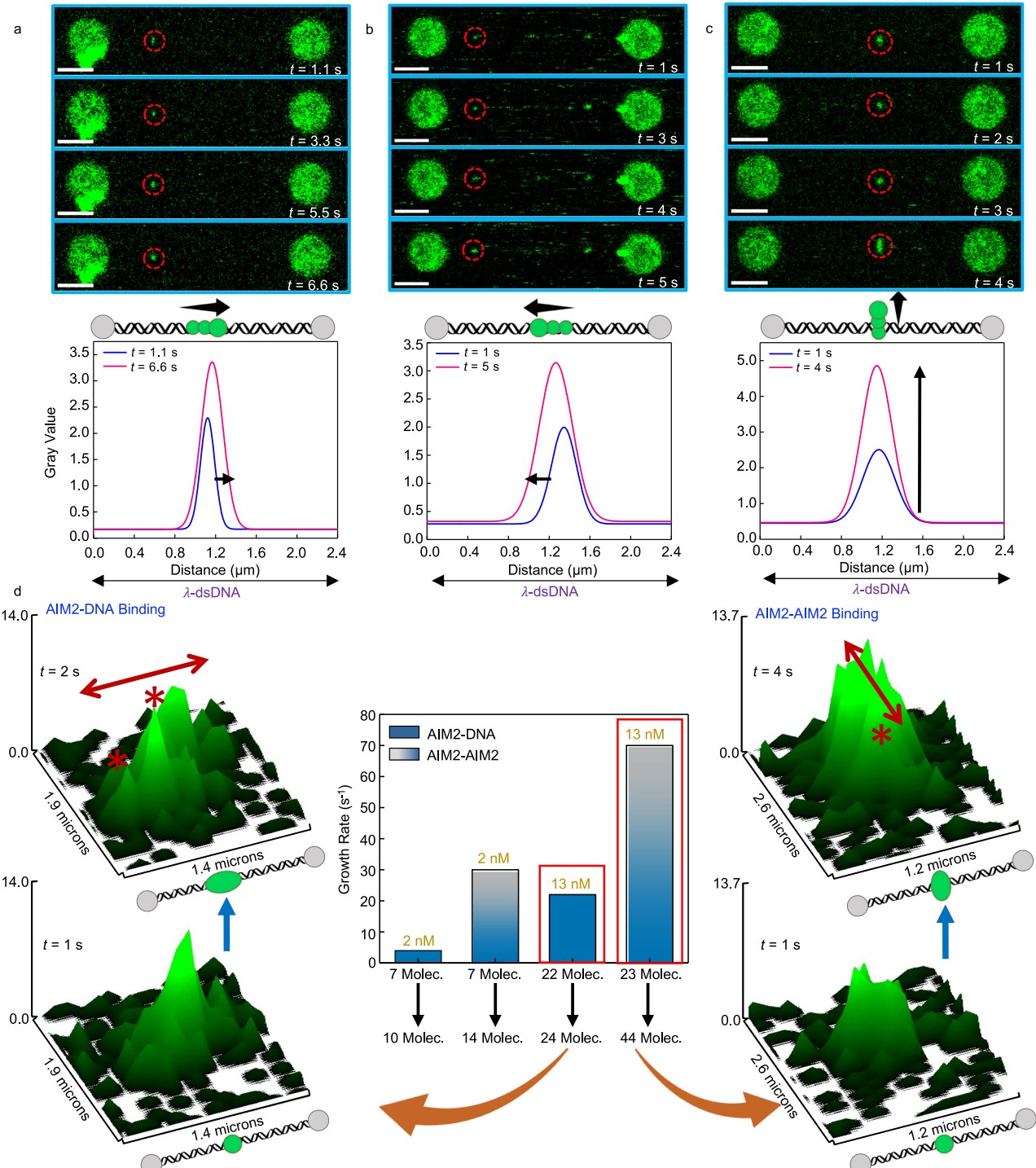

**Fig. 6 | AIM2 oligomers grow via two distinct mechanisms. a–c** Top: Representative confocal scans of movies (*n* = 45) showing AIM2 oligomer growth. Bottom: Oligomer growth represented by the increase in fluorescence intensity (gray value) to the right (**a**), left (**b**), and perpendicular to the DNA (**c**). Scale bars: 3 μm. **d** Center: Oligomer growth rates as changes in fluorescence intensity with time for clusters with different starting numbers of molecules (*X*-axis) at two AIM2 concentrations. Blue and blue-gray bars indicate oligomer growth via AIM2 binding to DNA and AIM2-AIM2 binding without DNA binding, respectively. Left and right: Fluorescence intensity plots at different times of AIM2 oligomers with 22 and 23 molecules. Growth via AIM2-DNA binding (left) and AIM2-AIM2 binding (right) show increased fluorescence along the DNA axis and the axis perpendicular to the DNA, respectively (asterisks and arrows). Molecule (Molec.). Source data are provided as a Source Data file.

incorporating the incoming AIM2 molecules independently of DNA binding explains the observation of asymmetric clusters (Figs. 1b, 3b).

In some cases, growth through PYD-PYD interaction is significantly faster than via DNA binding at the same protein

concentration and similar starting number of molecules. For example, Fig. 6d shows significantly different growth rates for two clusters with 22 and 23 molecules, respectively. The slower rate corresponds to a cluster with AIM2 incorporation along the DNA, as shown by the increase in fluorescence intensity in this direction. In contrast, AIM2

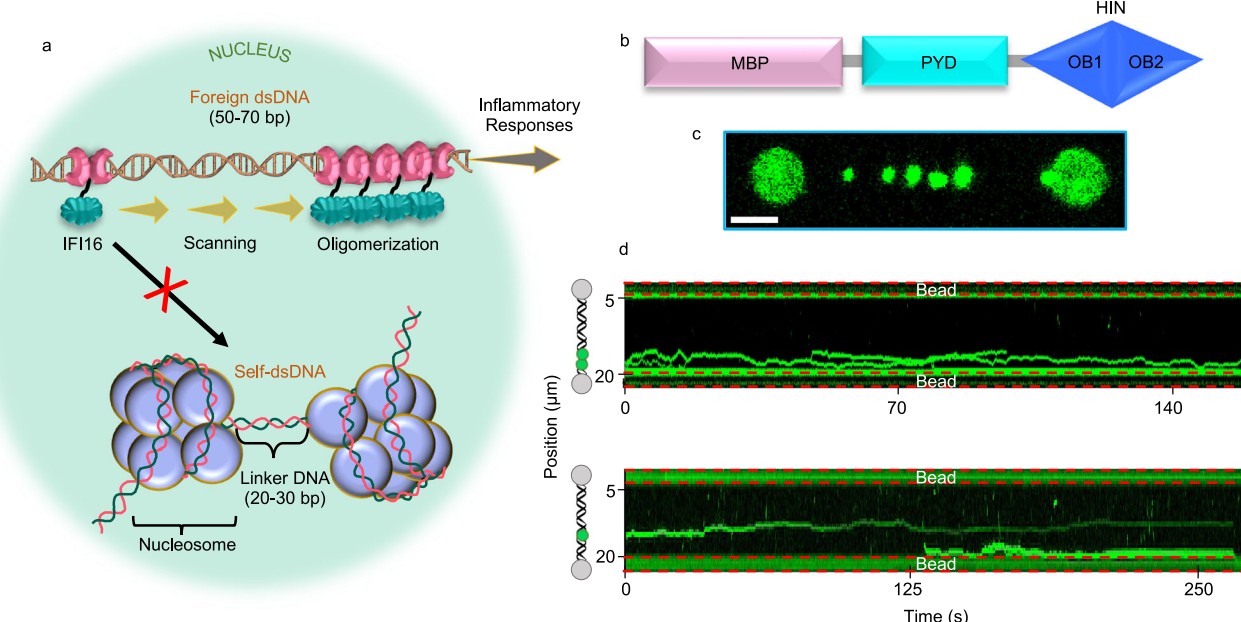

**Fig. 7 | MBP-AIM2 diffuses along the dsDNA analogously to IFI16. a** Schematic representation of IFI16 diffusion along the dsDNA as a mechanism to discriminate between self- and foreign-DNA[28]. **b** Domain organization of MBP-AIM2. **c** Representative two-dimensional scan (*n* = 70) of MBP-AIM2 oligomers bound to dsDNA (680 pM MBP-AIM2 and 160 mM KCl). **d** Representative kymographs (*n* = 40) of MBP-AIM2 (1 nM and 680 pM, top and bottom, respectively, and 160 mM KCl), showing binding and diffusion along the dsDNA. Scale bar represents 3 µm. MBP (Maltose Binding Protein), PYD (Pyrin Domain), HIN (Hematopoietic, Interferon-inducible, Nuclear localization), OB (Oligonucleotide/Oligosaccharide Binding).

molecules incorporate perpendicular to the DNA for the oligomer that grows faster, as indicated by the increase in fluorescence in this direction (Fig. 6d). This result could be attributed to the fact that cluster growth by AIM2-DNA binding requires two types of interactions; HIN-DNA and PYD-PYD, whereas only one type of interaction is needed for growth mediated by PYD-PYD binding.

In general, oligomer growth that is purely perpendicular to the dsDNA is less abundant compared to growth along the dsDNA. We have also observed a combination of the two types of growth with fluorescence intensity increasing perpendicularly and along the dsDNA. These growth mechanisms are further illustrated in Supplementary Fig. 10, which also includes cluster growth rates, as well as the starting and final number of molecules.

### AIM2 does not diffuse along the dsDNA molecule

The interaction between IFI16 and dsDNA has been studied previously by TIRF[28]. IFI16 molecules diffuse several µm along λ-phage dsDNA and find already bound IFI16 clusters for interaction[28] (Fig. 7a). Diffusion decreases and stops when clusters grow to ~8 molecules[28]. It was found that the minimum dsDNA length required for efficient IFI16 oligomerization is 50–70 bp. The presence of nucleosomes results in shorter dsDNA sequences, thus hindering IFI16 diffusion and self-assembly[28] (Fig. 7a). In contrast, low chromatinization of foreign dsDNA leads to the exposure of long stretches of dsDNA, thus allowing IFI16 to diffuse freely and oligomerize. Based on these results, the authors elegantly explain how IFI16 discriminates between host and foreign dsDNA[28].

We show here that AIM2 behaves differently as it does not diffuse along the dsDNA (Figs. 3b, c and 5). Both AIM2 clusters and single molecules do not significantly change position while attached to the dsDNA regardless of the length of time they are bound. Single AIM2 molecules show an average residence time of 3.3 s, whereas oligomers larger than 3 molecules show permanent attachment. We cannot rule out that diffusion of single molecules might occur at times longer than the average residence time. However, we have not observed motion of single AIM2 molecules on dsDNA even for the longer residence times

shown in Fig. 3. At the protein domain level, AIM2 contains only one dsDNA binding domain, whereas IFI16 bears two (Supplementary Fig. 1b, d). The presence of two HIN domains in IFI16 could explain the different diffusivity. In fact, the TIRF study reports that a truncated IFI16 construct lacking the PYD shows analogous diffusion behavior[28], which points to the key role of the HIN domains in the DNA binding mode.

The different behavior shown by AIM2 is intrinsic to the protein and not related to the experimental conditions. In fact, we have been able to observe diffusion along the dsDNA for a construct of AIM2 carrying the MBP (Maltose Binding Protein) tag (Fig. 7b–d) and Supplementary Movie 1). The capability of MBP-AIM2 to diffuse along the DNA must be related to the presence of the tag. It has been reported previously that the binding affinity of MBP-AIM2 for dsDNA is at least two orders of magnitude smaller than that of untagged AIM2 (Table 1), likely due to MBP interfering with the PYD-PYD driven oligomerization[27]. However, our results on the absence of diffusion of single AIM2 molecules do not include a PYD-PYD binding effect, thus indicating that the MBP tag could directly affect the interaction between the HIN domain and DNA, which could lead to diffusion. Binding affinity values of AIM2 constructs lacking the PYD are a better reference for our single molecule data because the latter are not affected by oligomerization via PYD. For instance, the reported $K_D$ of MBP-AIM2[HIN] for dsDNA is 2.7 times larger than that of AIM2[HIN] (lacking the PYD and the MBP tag) (Table 1)[27], which points to the effect of the MBP tag on the binding between the HIN domain and dsDNA. Overall, our results on the diffusion of MBP-AIM2 and the comparison with the previously reported $K_D$ values indicate that a stronger interaction between DNA and AIM2 likely explains the absence of diffusion.

### Discussion

Detailed studies on the operating mode of the inflammasome sensor AIM2 have revealed that dsDNA binding and protein oligomerization are connected and cooperative[27]. Compelling evidence support a mechanism in which the dsDNA acts as a platform for AIM2 oligomerization[25,27]. In addition, it has been shown that filaments

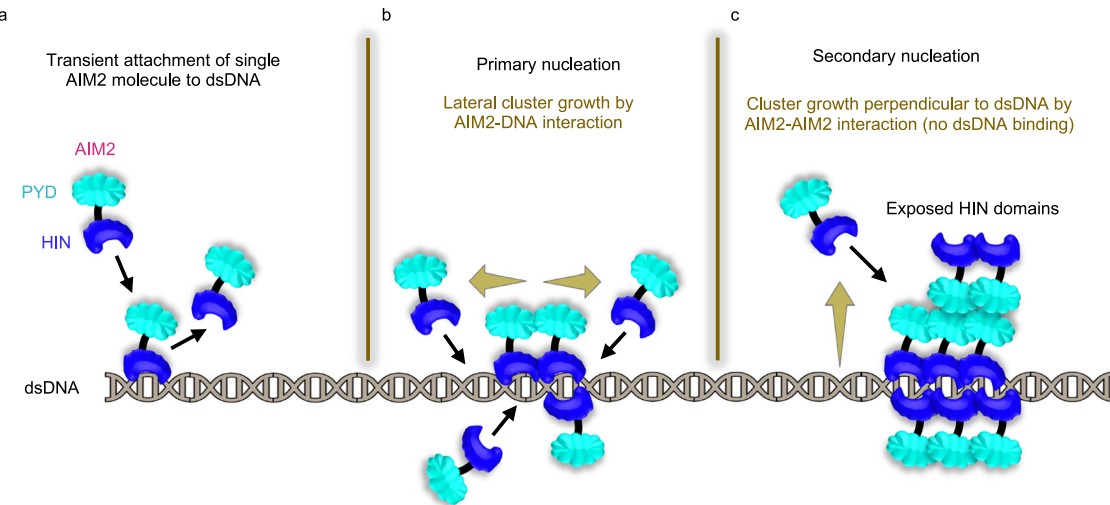

**Fig. 8 | Primary and secondary nucleation steps in the assembly mechanism of the AIM2 inflammasome sensor. a** Stochastic attachment of AIM2 to dsDNA and detachment. **b** Attachment of AIM2 molecules to prebound AIM2 oligomer via dsDNA binding, resulting in permanent attachment. Cluster grows along the dsDNA. **c** Attachment of AIM2 molecules to prebound AIM2 oligomer via AIM2-AIM2 interaction in the absence of dsDNA binding, resulting in permanent attachment. Cluster grows in direction perpendicular to dsDNA. PYD (Pyrin Domain), HIN (Hematopoietic, Interferon-inducible, Nuclear localization).

formed by AIM2 oligomerization upon dsDNA binding function as a template for the polymerization of the inflammasome adaptor ASC[27]. This mechanism helps explain the robust inflammatory response observed upon cell treatment with dsDNA fragments sufficiently long to trigger the formation of the AIM2 inflammasome[25].

We have presented here fundamental information on the kinetics and mechanism of propagation of AIM2-DNA assemblies to further our understanding of the initial stages of inflammasome formation at the molecular level. AIM2 oligomerizes into distinct, individual assemblies of different sizes coexisting on the same dsDNA molecule (Fig. 1b). Most small oligomers composed of 1–20 molecules grow with a rate ranging from 0.2–7 molecules/s at low nanomolar concentrations; however, we have observed growth rates up to 28 molecules/s. This time scale aligns with previously reported kinetic studies in live cells showing that the ASC speck assembles in approximately 3 min once ASC concentration redistributes upon AIM2 inflammasome activation[53]. The long-lived nature of the ASC speck is demonstrated in these studies by its persistence for several hours[53]. Our results on the permanent attachment of AIM2 oligomers are in accord with these observations.

Based on the association and dissociation rates of AIM2 and dsDNA at the single molecule level, we have determined the affinity of AIM2 for dsDNA devoid of self-association effects. The $K_D$ falls in the sub-micromolar range, likely due to the lack of sequence specificity for dsDNA. As expected, AIM2-DNA affinity is significantly low (approximately two orders of magnitude) compared to sequence-specific proteins such as transcription factors.

We have shown here that AIM2 oligomerizes to some extent in the absence of dsDNA at sub-nanomolar concentration. The use of single molecule techniques likely has facilitated the detection of these assemblies. AIM2 self-association unaccompanied by dsDNA binding could lead to sterile inflammation. A potential mechanism to tightly control AIM2 self-association might involve keeping a low basal concentration. In addition, it has been proposed that AIM2 self-association could be controlled by an autoinhibitory mechanism involving interactions between the PYD and HIN domains of AIM2[25]. Intramolecular interactions in AIM2 could compete with oligomerization and dsDNA binding at very low basal concentrations. Based on our results and in accordance with previous reports, we hypothesize that once pathogenic dsDNA enters the cytoplasm, AIM2 concentration increases, thus facilitating protein oligomerization and irreversible dsDNA binding,

which subsequently leads to active inflammasome formation and downstream signaling until the cell dies.

Finally, we have shown here that AIM2 does not diffuse along the dsDNA. Because only aberrant host dsDNA can be found in the cytosol, AIM2 might not follow the proposed mechanism for IFI16 to discriminate between host and foreign dsDNA, which involves diffusing on sufficiently long stretches of dsDNA to facilitate oligomerization[28]. We hypothesize that specific interactions of the HIN domains of AIM2 and IFI16 with dsDNA, as well as the presence of one versus two HIN domains, lead to different binding behavior. In fact, the reported binding affinity between full-length IFI16 and dsDNA is significantly smaller than that of AIM2 (Table 1)[26,27]. Another discrepancy in the behavior of AIM2 and IFI16 is that the HIN domain of AIM2 has been shown to oligomerize in the DNA[27], unlike the HIN domains of IFI16[26]. Moreover, the X-ray structures of the complexes between AIM2-HIN and IFI16-HINb with DNA revealed that the former leads to a larger solvent-accessible surface area buried upon complex formation[25]. Interdomain dynamics of IFI16 HIN domains, which are absent in AIM2, could play a role in diffusion. Additional studies are required to fully understand the different diffusion behavior of dsDNA sensors.

Based on the data reported here, we suggest a model for the formation of AIM2-DNA assemblies that involves three complementary and likely simultaneous scenarios (Fig. 8). Single AIM2 molecules stochastically and transiently bind to dsDNA with low affinity leading to survival times of approximately 3 s (Fig. 8a). During this time, incoming AIM2 molecules bind to dsDNA in positions that are sufficiently close to prebound AIM2 molecules or oligomers, causing lateral growth of the oligomer and permanent attachment (Fig. 8b). In conjunction or alternatively, incoming AIM2 molecules may also bind to prebound clusters via AIM2-AIM2 interactions (in the absence of dsDNA binding), leading to oligomer growth in direction perpendicular to the dsDNA (Fig. 8c). Overall, AIM2-DNA assembly formation and propagation can be understood as a polymerization process with primary (AIM2-DNA binding) and secondary (AIM2-AIM2) nucleation events. The double-nucleation mechanism is a well-known process demonstrated previously for sickle fiber formation by hemoglobin S polymerization[54]. The different propagation mechanisms proposed here (Fig. 8b, c) likely favor the formation of intertwined and densely packed filamentous structures via (1) AIM2[PYD]-AIM2[PYD] interactions between nucleoprotein filaments; (2) AIM2[HIN] interactions between nucleoprotein

filaments and free dsDNA fragments; and (3) AIM2$^{PYD}$-ASC$^{PYD}$ interactions between nucleoprotein filaments and ASC or polymerized ASC. This mechanism explains the random appearance of AIM2 clusters of different sizes and shapes (Figs. 1b, 3b), likely resulting from the favored binding of incoming AIM2 molecules to larger pre-existing bound oligomers. Additionally, attached AIM2 clusters of increasing size limit dsDNA availability, thus leading to a higher probability of incoming molecules to interact with prebound oligomers.

## Methods

Company names and catalog numbers of commercial reagents are provided as Supplementary Data 1.

### Synthesis and cloning of human AIM2

The full-length human AIM2 gene (amino acids 1-343) with an N-terminal 6xHis tag followed by a MBP tag and Tobacco Etch Virus protease (TEVp) recognition site (ENLYFQG) was synthesized and cloned into the pET21(b) vector by Gene Universal Inc. In addition, this construct includes a sortase A recognition site (LPETG) connected to the C-terminus of AIM2 by a flexible linker (GGGGS) and with two extra glycine residues after the recognition site to ensure optimal results of the sortase-mediated transpeptidation reaction[55] that is used to label AIM2 with a fluorophore-tagged peptide.

### Expression and purification of AIM2 constructs

The plasmid containing the AIM2 construct was transformed into Rosetta (DE3) cells, which were grown overnight in LB medium supplemented with 100 µg/mL ampicillin and 34 µg/mL chloramphenicol at 37 °C and 220 rpm. Overnight seed culture was transferred into large volume of LB medium and grown at 37 °C to an OD$_{600}$ of 0.6–0.8. Protein expression was induced at 18 °C with 1 mM isopropyl β-D-1-thiogalactopyranoside (IPTG) and incubated overnight. Cells were harvested and resuspended in lysis buffer containing 20 mM HEPES, pH 7.4, 400 mM KCl, 1 mM 2-mercaptoethanol (BME), 0.1% Triton X-100 and 5% glycerol. The cells suspension, supplemented with 1 mM phenylmethylsulfonyl fluoride (PMSF), 100 µg/mL lysozyme, and a protease inhibitor cocktail (Pierce protease inhibitor tablet contains AEBSF, aprotinin, bestatin, E-64, leupeptin, and pepstatin), was incubated at 4 °C for 30 min. Cells were lysed by at least 7 cycles of freeze-thaw using dry ice/ethanol bath and centrifuged at 126,224 g for 40 min. The supernatant was collected, and the pellet was washed again in lysis buffer. Supernatants were collected after each centrifugation step. To remove the bacterial DNA, a 2–3% solution of streptomycin sulfate was added dropwise into the supernatant, followed by constant stirring at 4 °C for 30 min. The insoluble precipitates were separated by centrifugation at 126,224 g for 40 min. The streptomycin sulfate precipitation step was repeated twice to achieve efficient elimination of DNA.

The supernatant obtained after centrifugation was filtered with a 0.45 µm pore filter and applied onto a 5 mL Ni-NTA column (Thermo Fisher Scientific) preequilibrated with lysis buffer. The column was washed in two steps with lysis buffer containing 25 mM and 50 mM imidazole (50 mL each). MBP-AIM2 was eluted in lysis buffer containing 400 mM imidazole. Fractions were collected and analyzed by SDS–PAGE. Fractions containing MBP-AIM2 were pooled and purified by ion-exchange chromatography using 5 mL HiTrap-SP column (GE Healthcare) equilibrated in 20 mM HEPES, pH 7.4, 160 mM KCl, 1 mM BME, 0.1% Triton X-100, and 5% glycerol on an HPLC system. After washing, the protein was eluted with a linear gradient ranging from 0.16 M to 1 M KCl at a flow rate of 1 mL/min and subsequently analyzed for purity by SDS–PAGE. Protein concentration was determined by UV absorption measured at 280 nm using a molar extinction coefficient ($\varepsilon$) of 75,290 M$^{-1}$ cm$^{-1}$ for MBP-AIM2. The A$_{260}$/A$_{280}$ ratio of the different fractions was measured to identify DNA-free protein, and the

fractions with a ratio of ~0.64 were pooled. Dialysis and concentration steps of protein solutions were avoided throughout the purification process to reduce protein oligomerization.

To remove the MBP tag, 450 nM MBP-AIM2 was mixed with 18 µM TEVp in a buffer containing 20 mM HEPES, pH 7.5, 160 mM KCl, 1 mM BME, 0.1% Triton X-100, and 5% glycerol and incubated at 30 °C, 220 rpm for 1 h. The reaction was subjected to centrifugation to separate protein precipitation due to MBP removal, and the supernatant was further used for fluorescent labeling. For experiments requiring the MBP-AIM2 construct, the TEVp cleavage step was omitted.

### Labeling of AIM2 constructs with Alexa Fluor 488

To ensure single fluorophore labeling of AIM2 and avoid mutations commonly used in fluorophore labeling strategies, a short peptide conjugated with Alexa Fluor 488 was attached covalently to the protein constructs using sortase A transpeptidation.

**Labeling of the short peptide with Alexa Fluor 488 maleimide dye.** A peptide with amino acid sequence, GGGC, was synthesized and purified by Thermo Fisher Scientific. The peptide was dissolved in a labeling buffer containing 20 mM sodium phosphate, pH 7.2, and 100 mM NaCl. Alexa Fluor 488 C$_5$ maleimide (Thermo Fisher Scientific) was dissolved in DMSO. The labeling reaction was prepared by mixing a 300 µM peptide solution and Alexa 488 at a 2-fold molar excess relative to the peptide concentration. The reaction mixture was incubated at 25 °C for 4 h at 220 rpm. The reaction was quenched by adding BME. The labeled peptide was purified by reverse-phase chromatography using ZORBAX 300SB-C18 column (Agilent) equilibrated with 5% acetonitrile, 94.9% H$_2$O, and 0.1% TFA and eluted in a gradient created with a buffer containing 5% H$_2$O, 94.9% acetonitrile and 0.1% TFA. The eluted fractions were collected.

Mass spectrometry was used to determine the percentage of labeled peptide. The experimental mass obtained for the unlabeled peptide (292 g/mol) matches the expected theoretical weight (292 g/mol). The experimental mass obtained for Alexa 488 was 699 g/mol. Mass spectra acquired after labeling the peptide with Alexa 488 showed a major peak at 991 g/mol, in agreement with the expected molecular weight. In addition, we observed a peak at 287 g/mol at percentages ranging from 21% to 1%, but typically closer to 10% or smaller. We attributed the presence of this peak to the unlabeled peptide. Based on these data, we estimated an average percentage of labeled peptide of 90 ± 8%.

The concentration of the labeled product was determined from the absorbance at 493 nm using a molar coefficient of 72,000 M$^{-1}$ cm$^{-1}$ for Alexa 488. The labeled peptide was lyophilized and stored at −80 °C for further use.

**Pairing peptide-Alexa 488 conjugate with MBP-AIM2 and AIM2.** The covalent attachment of the GGGC peptide-Alexa 488 conjugate with the LPETG-containing MBP-AIM2 and AIM2 (without MBP tag) was performed by sortase-mediated C-terminal transpeptidation.

To label MBP-AIM2, the protein and the peptide conjugate were mixed at a 1:4 molar ratio. Specifically, solutions of MBP-AIM2 and peptide-Alexa 488 conjugate were mixed in sortase reaction buffer (20 mM HEPES, pH 7.5, 160 mM KCl, 1 mM BME, 0.1% Triton X-100, and 5% glycerol, and 10 mM CaCl$_2$), resulting in final protein and peptide concentrations of ~20 µM and 80 µM, respectively. Sortase A was added to a final concentration of 10 µM, and the reaction was incubated overnight in the dark at 4 °C. Subsequently, MBP-AIM2 labeled with peptide-Alexa 488 was purified by size exclusion chromatography (SEC) using Superdex 200 increase 10/300 GL column (GE Healthcare) in a buffer containing 20 mM HEPES, pH 7.5, 160 mM KCl, 1 mM BME, 0.1% Triton X-100 and 5% glycerol buffer at a flow rate of 0.5 mL/min. The concentration of the labeled fraction was determined from the

absorbance values ($A$) at 280 nm and 493 nm by considering the dye's absorbance at 280 nm using a correction factor ($CF$) indicated by the manufacturer of 0.11. A theoretical molar extinction coefficient ($\varepsilon$) of 75,290 $M^{-1}$ $cm^{-1}$ was used for MBP-AIM2 to determine the protein concentration using the equation provided by the manufacturer:

$$\text{Protein concentration} = ((A_{280} - (A_{493} \times CF))/\varepsilon) \times \text{Dilution factor} \quad (1)$$

Mass spectrometry data of the purified labeled protein after the transpeptidation reaction shows a major signal at the expected molecular weight and no signal is observed for the unlabeled protein (Supplementary Fig. 8c). Based on these results, we estimate a sortase transpeptidation yield close to 100% and thus a protein labeling efficiency of 90% due to the labeling of the fluorescent peptide.

To label AIM2 (without MBP), the protein and the peptide conjugate were mixed in the sortase reaction buffer, resulting in final concentrations of 450 nM and 5 µM, respectively. Sortase was added to a final concentration of 10 µM. The reaction was incubated overnight in the dark at 4 °C. The purification protocol was identical to that of MBP-AIM2 (above). Fractions corresponding to monomeric AIM2 were pooled. The concentration of SEC fractions was determined by fluorescence spectroscopy at 25 °C using a Horiba PTI QuantaMaster 400 fluorimeter with a slit width of 10 nm for both excitation and emission. Briefly, fluorescence emission spectra of the labeled protein fractions were collected from 490 nm to 700 nm by exciting Alexa 488 at 480 nm. Maximum emission intensity was obtained at 519 nm. The concentration of the AIM2-peptide-Alexa 488 samples was calculated by extrapolating from a calibration curve based on the fluorescence emission of standard samples prepared from the MBP-AIM2-peptide-Alexa 488 in a concentration range from 0.5 nM to 10 nM.

## Single molecule experiments

Single molecule experiments were performed at room temperature on the C-Trap B08 system (LUMICKS) integrated with optical tweezers, confocal fluorescence microscopy and microfluidics, and recorded using BlueLake software version 2.3 (LUMICKS). Experiments were performed in freshly prepared imaging buffer containing 20 mM HEPES, pH 7.5, 160 mM KCl, 1 mM BME, 0.1% Triton X-100, and 5% glycerol supplemented with an oxygen scavenging system (5 mg/mL D-glucose, 20 µg/mL catalase and 100 µg/mL glucose oxidase) and 1 mM Trolox methyl ether filtered with 0.05 µm syringe filter. All the components (beads, DNA, and protein) were prepared in this buffer. A commercial five-channel laminar flow cell (LUMICKS) (Fig. 1a) mounted on an automated XY stage consists of 3 parallel channels (channels 1–3) separated by laminar flow and two orthogonal channels (channels 4 and 5). The flow cell was flushed with buffer ~20 min prior to sample addition. Streptavidin-coated polystyrene beads (Spherotech, 0.5% w/v stock) with a diameter of 3.11 µm were diluted 4/1000 in imaging buffer and placed in channel 1. Channel 2 was filled with biotinylated λ-phage dsDNA (48.5 kbp, LUMICKS) at a concentration of 24 pg/µL. Imaging buffer was flowing in channels 3 and 5, and fluorophore-labeled protein diluted in imaging buffer to the required working concentration was placed in channel 4. The typical flow rate was fixed at a pressure of 0.3 bar for all channels except for channel 4 where flow was off for image acquisition.

Two streptavidin-coated beads in channel 1 were optically trapped with a stiffness of ~0.4 pN/nm using 1064 nm trapping lasers. One molecule of λ-phage dsDNA was tethered between the two beads by moving the traps from channel 1 to channel 2 that are subjected to laminar flow. The traps were moved to channel 3 (buffer), where the presence of a single molecule of dsDNA was verified by comparison of the experimental force-extension curve to the built-in Worm-like chain (WLC) model in the BlueLake software. The bead-DNA complex was then moved to protein channel 4 and incubated for protein-DNA binding. Flow from channel 4 was kept off during data acquisition. For

confocal imaging, fluorophore labeled AIM2 was excited at 488 nm and emission was detected using a 512/25 nm bandpass filter. Confocal two-dimensional (2D) images were recorded in single and continuous scanning modes. Kymographs were generated via a confocal line scan through the center of the two beads along the dsDNA molecule in continuous mode. Other imaging conditions included 10% laser power, 50 µs/pixel time, and 100 nm pixel size.

## Flow cell passivation

The protein channel in the flow cell was passivated by flowing a 0.1% (w/v) solution of BSA followed by a 0.5% pluronics solution at 0.3 bar for 20 min each. In addition, BSA at 0.2 mg/mL was added to all buffers used in the C-Trap. Passivation was used for results shown in Fig. 5 and Supplementary Fig. 10 on AIM2 oligomer growth. The presence of BSA does not change the protein behavior as similar results were obtained in the absence of BSA.

## Processing of confocal images and kymographs

Confocal 2D images and kymographs were exported from BlueLake in HDF5 file format and processed using custom-written scripts in the Pylake Python package provided by LUMICKS. 2D images converted into TIFF format were further analyzed using Fiji (Image J) for cluster size and growth analysis. In Fiji, the RGB images were split to retain data from the blue channel, converted into green for clear visualization, and the color contrast level was adjusted. A custom-written kymotracking script, available at the script sharing platform Harbor (LUMICKS), was used to track the binding traces in the kymographs and extract information on time, photon counts, and position of individual protein molecule/oligomer on the DNA[56–58].

Kymograph heat maps were generated using a custom Python script. Firstly, kymographs in HDF5 format were converted into RGB TIFF images, and the blue channel was extracted. A median filter was applied to reduce the background noise. The color contrast and brightness of individual images were adjusted to enhance trace visibility as indicated by the color bar (Fig. 5). Finally, heat maps were generated by using the 'plasma' colormap. The left panels of Fig. 5 were created by extracting the blue channel from the RGB images and converting them into green for clear visualization using Fiji. The color contrast was adjusted for each kymograph image. To compute the number of protein molecules indicated in the heat maps, two $7 \times 7$ pixel areas were selected at the initial and maximum intensity points of the corresponding traces. The average fluorescence intensities of these areas were determined to quantify the number of molecules prior to color contrast or brightness adjustment.

## Single step photobleaching analysis

Kymograph binding traces representing AIM2 oligomers bound to dsDNA were tracked as described above. These trajectories represent changes in fluorescence intensity (photon counts) as a function of time. In many instances, a decay in fluorescence intensity is observed due to the photobleaching of fluorescent AIM2 oligomers exposed to the excitation laser, thus resulting in complex trajectories because of the large number of fluorophores. In such cases, visual inspection is necessary to identify single and multiple photobleaching steps by calculating the average intensity of flat regions before and after observed intensity drops (Supplementary Fig. 4a). In the analysis of these trajectories, the smallest steps were considered single photobleaching events ($n = 48$), which were binned with a width of 3 photon counts to construct a histogram that was fit to a Gaussian function using Origin (OriginLab 2017). Fitting reveals an average of $11 \pm 4$ photon counts for a single fluorophore (Supplementary Fig. 4b). This result was used to calculate the number of molecules forming AIM2 clusters.

To increase confidence in the visual analysis, representative trajectories were analyzed with an automatic method for step detection

known as "AutoStepfinder[59]". This method finds the same number of steps we detected by visual inspection for several representative trajectories. Specifically, for the trajectory shown in Supplementary Fig. 4a, "Autostepfinder" can detect 2 or 3 steps with identical quality of fit (Supplementary Fig. 5a, b). Our visual inspection indicates 3 steps. In another representative example, "AutoStepfinder" detects 2 steps in the raw data, matching the results from visual inspection (Supplementary Fig. 5c). In addition, the sizes of the steps are in agreement with the visual analysis (Supplementary Fig. 5).

### Analysis of photon counts from single fluorophore bound to dsDNA

To confirm the results from the photobleaching analysis, we determined the photons emitted by a molecule carrying a single fluorophore. This molecule is biotinylated λ-dsDNA to which a single fluorophore (Atto 488) is attached at base pair position 33,786 (Supplementary Fig. 6). This molecule, synthesized by LUMICKS, is a gift from the laboratory of Prof. Muñoz (UC Merced). The analysis of 23 traces observed in kymographs resulting from the fluorescence of the single Atto 488 bound to dsDNA shows an average emission of $12 \pm 2$ photons (Supplementary Fig. 6c). This value is very close to the 11 photon counts determined by the photobleaching step analysis. It is reasonable to consider that Alexa 488 and Atto 488 emit a similar number of photons as both fluorophores have very similar molecular structures[43,45], (Supplementary Fig. 6d), similar quantum yields (Alexa 488: 0.92; Atto 488: 0.80) and identical lifetimes (4.1 ns)[44,46]. These experiments were performed with the same photoprotection cocktail used for AIM2.

### Photon count underestimation due to detector dead time

The avalanche photodiode detector model and manufacturer name are LUMICKS' proprietary information. Information related to the detector of the C-Trap B08 model shared by LUMICKS is shown below.

- Detector dead time → 35 ns
- Photon detection efficiency (PDE) → at 650 nm, 75%; at 830 nm 50%, at 512 nm, 50–70%

To estimate photon count underestimation due to the detector dead time, we have followed an idealized dead time model[31] that assumes the detector is not affected by events happening during the dead time, "$\tau$" ($\tau = 35$ ns). This implies the detector is dead for a fixed time after each event[31].

For a counting rate "$m$," the fraction of time during which the detector is dead is "$m\tau$." The fraction of time during which the detector is detecting is "$1 - m\tau$." The number of true events per unit of time is "$n$."

Thus, the number of true events that can be detected is:

$$n = \frac{m}{1 - m\tau} \tag{2}$$

We have used a pixel dwell time of 50 μs in which we detect 11 photons per fluorophore, which gives a detection rate of $220 \cdot 10^3$ photons/s.

In this case, the real photon count is:

$$n = \frac{220 \cdot 10^3}{1 - 220 \cdot 10^3 \cdot 35 \cdot 10^{-9}} = 221,707 \text{ photons/s} \tag{3}$$

Thus, there is an underestimation of 0.8% for one fluorophore.

If the cluster has 10 fluorophores, the detection rate will be $2200 \cdot 10^3$ photons/s. However, the real number of photons/s will be:

$$n = \frac{2200 \cdot 10^3}{1 - 2200 \cdot 10^3 \cdot 35 \cdot 10^{-9}} = 2,383,532 \text{ photons/s} \tag{4}$$

Thus, resulting in an underestimation of 7.7% for a cluster of 10 fluorophores. This percentage leads to approximately one fluorophore not being counted in a cluster of 10.

### Point spread function analysis of the C-Trap's confocal microscope

2D images of commercial fluorescent beads of 23 nm (Beads RGB, GATTAquant) and 100 nm (TetraSpeck, Thermo Scientific) in diameter were obtained at a resolution of 30 nm per pixel by exciting at 638 nm (23 nm bead) and 488 nm (100 nm bead), respectively. The images were analyzed with Fiji. The corresponding PSFs of the 100 and 23 nm beads were analyzed by LUMICKS (proprietary software) and us (Fiji and qtGrace for fitting), respectively, providing FWHM in the X and Y axes shown in Supplementary Fig. 2.

### Cluster size distribution and cluster growth analysis

2D confocal images were analyzed with Fiji to determine the number of molecules and real-time growth of AIM2 clusters (oligomers) bound to λ-phage dsDNA. Individual clusters were encompassed using Fiji's selection tool, and the obtained fluorescence intensity was plotted with Origin (Fig. 2). Intensity data were corrected by subtracting the average background intensity for the same area of the selection tool applied to 3–5 different regions of the 2D scans. The resulting intensity values were divided by 11 (i.e., the number of photons emitted by a single fluorophore) and further corrected for labeling efficiency and emission at red wavelengths (Supplementary Fig. 3) to obtain the cluster sizes.

To determine cluster surface areas, individual clusters and the surrounding background were encompassed using Fiji's rectangular selection tool. The profile of the fluorescence intensity was fitted to a Gaussian function. The area, assuming the cluster adopts a circular shape, was determined using the Gaussian's Full Width at Half Maximum (FWHM) as the cluster's diameter. A histogram depicting the surface areas of 106 clusters was constructed with a bin size of $0.05 \ \mu m^2$ using Origin.

For real-time cluster growth analysis, 2D confocal images were captured in continuous scanning mode and converted into individual, time-stamped, sequential frames using LUMICKS Pylake python package. The increase in intensity of individual frames was analyzed in Fiji and plotted as a function of time in Origin (Fig. 4 and Supplementary Fig. 10). Cluster growth direction was determined using Fiji by obtaining profile plots of mean gray values as a function of distance along the DNA axis. The direction of intensity increase matches the direction of cluster growth (Fig. 6a–c). To generate movies, individual frames were combined at a rate of two frames per second (Supplementary Movie 1).

Oligomer growth rates shown in Fig. 6d (center) and Supplementary Fig. 10 were determined as the difference between photon counts of final and initial frames divided by the total observation time. For this analysis, 2D frames of 4 clusters were used, including 2 clusters starting with 7 molecules at 2 nM AIM2, and 2 clusters starting with 22 and 23 molecules at 13 nM AIM2 (Fig. 6d, center). The three-dimensional plots shown in Fig. 6d (left and right), and Supplementary Fig. 10 were generated with Fiji from the analysis of the corresponding clusters.

### Force dependent measurements

The λ-phage dsDNA molecule tethered between the two trapped beads was stretched to 16 μm and 16.5 μm at forces of 17 pN and 40 pN, respectively. All data were acquired at 17 pN except for data shown in Supplementary Fig. 9, which were obtained at 40 pN. The C-Trap instrument uses bright field imaging and template recognition to determine inter-bead distance. Laser interferometry is used for force measurements.

## Dwell time and kinetic analysis

We tracked single molecule binding traces from kymographs to extract photon counts and residence time (dwell times) information (Fig. 3). The histogram of dwell times of 314 single molecule traces using a bin width of 1 s, appropriate for the data set[60], included a first bin with only 20 traces (6% of the total data set). Thus, this first bin was removed to obtain the dissociation rate constant ($k_{off}$) by fitting the resulting histogram to a single exponential decay function (Eq. 5) using Origin (Fig. 3d). We analyzed 100 and 109 single molecule traces at 1 nM and 5 nM AIM2 concentrations, respectively, to determine the effect of protein concentration on the $k_{off}$. The data at 1 nM showed a first bin with lower number of traces (16% of the total data set) and was removed. Histograms with 1 s bin width were fitted to a single exponential decay function to determine $k_{off}$ values at the two concentration values (Fig. 3e, f). The $k_{off}$ value at 40 pN was determined by analogous fitting to a single exponential of the corresponding dwell time histogram ($n = 172$) (Supplementary Fig. 9).

$$y = A \exp^{(-k_{off} \cdot t)} + y_0 \tag{5}$$

Where $y$ represents the number of binding traces, $y_O$ is the exponential baseline and $t$ is the residence time.

Association rate constants ($k_{on}$) were calculated based on the analysis of 100 single molecule traces at 1 nM and 109 single molecule traces at 5 nM in kymographs acquired with a constant time length of 600 s. The unbound times ($t_{on}$) were calculated from the total observation time minus the sum of the residence time of all the traces for each kymograph. The obtained values of $t_{on}$ are summarized in Supplementary Tables 2 and 3. The association rates (Eq. 6) were calculated using the average $t_{on}$ at 1 nM and 5 nM concentrations ($C$). Finally, the dissociation constant ($K_D$) was determined from the ratio of the dissociation rate ($k_{off}$) and association rate ($k_{on}$). These results are shown in Table 1.

$$k_{on} = \frac{1}{[C] \cdot t_{on}} \tag{6}$$

## Mass spectrometry

The purity and integrity of the recombinant proteins and peptides were determined by mass spectrometry and SDS-PAGE. Lyophilized peptide material was diluted with 50% acetonitrile and directly injected to an electrospray ionization mass spectrometer (Q-Exactive Hybrid Quadrupole-Orbitrap, Thermo Scientific). Protein solutions were diluted with a solution containing 95% acetonitrile, 4.9% water, 0.1% formic acid and injected to reversed phase column (Acclaim 200 C18, 3 μm, Thermo Scientific operating at a flow rate of 0.3 mL/min) for subsequent mass spectrometer analysis. The molecular weight obtained matched the expected molecular weight based on the amino acid sequences.

## Data analysis

Confocal images and kymographs were processed and analyzed with the Pylake software (https://lumicks-pylake.readthedocs.io/en/stable) version 0.13.2 and with scripts retrieved from the Harbor platform (https://harbor.lumicks.com).

Intensity and area in confocal images were further analyzed with Fiji (https://imagej.net/software/fiji/downloads) version 1.0 or ImageJ2 version 2.9.0.

Graphs were plotted and analyzed using OriginLab (https://www.originlab.com) version 2017 and qtGrace (https://sourceforge.net/projects/qtgrace) version 0.2.6.

## Reporting summary

Further information on research design is available in the Nature Portfolio Reporting Summary linked to this article.

## Data availability

All data are available within this manuscript and its supplementary information files. Original files in "HDF5" format corresponding to 2D confocal scans and kymographs are available from the corresponding author upon request. Source data are provided with this paper.

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

## Acknowledgements

Research reported in this publication was supported by the National Institute of Allergy and Infectious Diseases of the National Institutes of

Health under award number R21AI168983 to E.d.A. The content is solely the responsibility of the authors and does not necessarily represent the official views of the National Institutes of Health. E.d.A. and M.S. acknowledge support from the National Science Foundation CREST Center for Cellular and Biomolecular Machines (CCBM) under award number NSF-HRD-2112675. We are grateful to Dr. Mourad Sadqi (UC Merced, CCBM) for mass spectrometry data acquisition and analysis, Luis I. Gutiérrez (University of Granada, Spain) for TEM image acquisition, and Dr. Pawan Panwar (Milwaukee School of Engineering, Wisconsin) for support on Python and MATLAB programming. We thank UC Merced Imaging and Microscopy Facility.

## Author contributions

E.d.A. conceived the project. M.S. and E.d.A. designed the experiments. M.S. prepared the samples, performed the experiments, and acquired the data. M.S. and E.d.A. analyzed the data. M.S. and E.d.A. wrote the manuscript.

## Competing interests

The authors declare no competing interests.
