## [Peer Review File · Nature Communications]

Assembly mechanism of the inflammasome sensor AIM2 revealed by single molecule analysisEditorial Note: Parts of this Peer Review File have been redacted as indicated to maintain the confidentiality of unpublished data, and to remove third-party material where no permission to publish could be obtained.

REVIEWER COMMENTS

Reviewer #1 (Remarks to the Author):

This manuscript by Sharma and Alba uses correlative optical traps and confocal fluorescence microscopy to investigate the assembly of AIM2 on DNA using single molecular techniques. Distinct mechanisms for AIM2 oligomerization via AIM2 binding to adjacent DNA or AIM2-AIM2 interactions are reported, likely through stochastic mechanisms. The k-on, k-off, and K_d values were obtained at the single-molecule level for AIM2-DNA interaction for the first time, confirming that PYD-PYD mediated association significantly enhances AIM2 binding to DNA. Furthermore, AIM2 does not seem to move along DNA as shown for IFI16, and MBP tag used previously may impact HIN-DNA interaction besides PYD-PYD association. The kinetics of AIM2-DNA binding and the accompanied AIM2 oligomerization at single-molecule level are interesting findings that should be of interest to the readership of Nature Communications.

Major issues:

The kymographs in the manuscript do not support the growing of DNA-bound AIM2 oligomers which calls into question the major findings of this manuscript. Figure 2b and statement in the text show that within seconds and at AIM2 concentrations of 0.2-1.4 nM, fluorescent AIM2 clusters of various sizes appeared on DNA. However, the kymographs in figures 4b and 7b do not show any increase of fluorescence intensities for single molecule trace up to a few minutes. In fact, the fluorescence intensity actually decreased or disappeared within that time frame perhaps from AIM2 dissociation or photobleaching. This inconsistency coupled with concern about the potential aggregation of the AIM2 sample prior to DNA binding (see below), suggest that maybe some of the clusters shown in figure 2b were pre-formed before DNA binding, rather than the AIM2 clusters growing on DNA.

Supple figure 2c shows AIM2 cluster/aggregates in the absence of DNA, which is concerning, especially since supplementary figure 3A does not show chromatogram earlier than 30 minute elution time that may reveal larger or aggregated AIM2 samples. A small fraction of aggregated or sticky AIM2 samples that may non-specifically associate with beads could skew the interpretation of AIM2-AIM2 interaction in cluster assembly. The flow cells or buffers may need to include BSA to reduce non-specific interactions and lend confidence to the observed data.

Minor issues:

Figure 1 shows signaling pathways and multiple published structures. Those are not part of the current work and may be more appropriate as a supplementary figure or for a review manuscript.

Figure 2c shows that the dominant AIM2 clusters are in 0.1-1 μ m² range, which may span a few thousand bp of B form dsDNA. This seems inconsistent with the estimate of 25 AIM2 molecules per cluster given the size of the dsDNA that may bind hundreds of AIM2 molecules.

Figures 5-6 show that the cluster size grew by 3-4 fold within a few seconds, which is inconsistent with the 13-15-molecule size of oligomers shown in figure 3 if the growth starts with 1-2 AIM2 molecules. Suggest increase the time frame to see if the cluster size may increase to the estimated dominant species of 13-15 oligomers.

Figure 6d shows t=1s and t=2s on the left, whereas t=1s and t=4s on the right. Such comparison may not accurately represent how AIM2-DNA binding compare with AIM2-AIM2 binding. For consistency, the authors should show t=4s instead of t=2s on the left.

Supple figure 1a shows 2-step photodepletion followed by 2 1-step photodepletion for a total of 4-step. However, because the photon counts are so scattered, a total of 3 1-step photodepletion seems to fit the data just as well as the 2-step+1-step+1step photodepletion. This will change the estimate of the average number of photons per single fluorophore in supple figure 1b, and the estimate of the number of AIM2 in each cluster. Suggest using photobleaching method to estimate single AIM2 photon count.

Since optical traps were used, it should be clarified whether the measurements on AIM2-DNA association observed here are dependent on the amount of force applied or not.

Scale bars are missing for many of the figures.

Reviewer #2 (Remarks to the Author):

This paper deals with the assembly process of a key component in the innate immune response, i.e. the AIM2 inflammasome sensor. This is a cluster of proteins that assembles around a foreign DNA and signals its existence to other systems in the cell. The authors use single-molecule spectroscopy of labeled AIM2 proteins to observe the formation of clusters around lambda-phage DNA molecule. They use the signals to count the number of protein molecules in the clusters and measure on and off rates. They also obtain information about the growth mechanism of DNA-bound oligomers. Somewhat surprisingly, they find that the protein does not diffuse on DNA.

The paper is interesting, but reading it raises multiple questions, as follows:

- Most importantly, I wonder about the counting method the authors use. My main concern actually arose after observing the growth experiments of Figure 6. In particular, Figure 6c shows an elongated cluster that from the analysis seems to be significantly larger than 0.5 micron. Given the size of a 324-residue protein like AIM2 (perhaps 2-3 nm), such a cluster should contain hundreds of protein molecules. Yet the authors conclude that it contains only ~20 molecules! The counting method should therefore be revised and also explained better.
 - In relation to this, the assignment of steps in sup. figure 1 is not so obvious- was any analytical method used for this assignment? How was it determined that some steps are due to two molecules, for example? This is crucial for the counting analysis.
 - As another related point, surely the assignment of 11 photons for the signal of a single molecule is just approximate. This means that there is an error that propagates into each determination of number of molecules- this error should be estimated and used for each calculation.
 - In this respect, one wonders whether fluorescence is quenched in clusters, leading to under-counting. There is some discussion of this point in the paper, but I did not see a conclusive test of this possibility.
 - The disappearance of fluorescent signals, as well as steps in signals, can be attributed to either photobleaching or dissociation of proteins. I am not sure I see a clear way to distinguish between the two in the paper.
 - The very large clusters reported in Supp Figure 2 are not well characterized and it is not clear when they appear and how their appearance might affect the results of the work.
- Additional point by order of appearance in the paper:
- It is not clear whether the cluster size distribution is fixed or time-dependent. Further, it seems from Figure 2C that small clusters might be missed by the analysis.
 - The specific detectors used in the experiment should be mentioned and how the authors have arrived at a counting underestimation of 6% should be explained. Photon counting vs. molecular number can in principle be tested and calibrated independently with other molecules.
 - Page 5, top paragraph- it is not clear what sigmoidal response is being discussed here and why.
 - Terminology in the third paragraph of the same page: the authors do not perform any 'correlative' microscopy here; it is not clear what 'genomic position' means.
 - Page 6, second paragraph: 'we can assume that the koff is zero for the longest DNA'- why should it be zero?
 - It is worrisome that the KD obtained here is so much larger than estimated by others. This could indicate that kon is underestimated significantly.
 - Page 7- why are the growth rates not linear with concentration? When the concentration is increased four-fold and then almost ten-fold, the growth rate increases only by a factor of less than 2 and then by a factor of 5, respectively.
 - Page 10- 'AIM2 does not need to diffuse'- proteins diffuse because of their interactions, not because they need or do not need. The question arises as to what makes AIM2 static when on the DNA, even though its interaction with the DNA is not that strong. This is surprising and requires further explanation.
 - Details about the microfluidic system are missing- is it home-built? Commercial? What prevents material from diffusing from channel 2 to the other channels? What is the flow rate in the other channels? What is the depth of the device? Is the whole depth being imaged?
 - Have the authors assessed the labeling efficiency? This is very important in their estimates of cluster sizes, as multiple unlabeled proteins can lead to significant underestimated counting.
 - Page 20- Is it important to show molecular structure of the proteins studied here?
 - Page 21- size of scale bars of Fig. 2b is missing. Please check all figures.

Reviewer #3 (Remarks to the Author):

This paper reports on single-molecule measurements of the assembly of AIM2 on dsDNA, a step in an inflammatory response. The study provides dynamic information on this assembly process, adding to existing biochemical and structural work. Overall, I found the aim of the experiments interesting and appropriate for a journal like Nature Communications. However, some of the work is focused on initial characterization, and while they are important, they are more an opening to other considerations that are missing, in my opinion. More specifically:

- 1) The large AIM2 assemblies are identified as clusters, and the language generally indicates that the authors think of them as static oligomers. While I agree with the conclusions that the larger clusters remain bound, and hence are static in that sense, the clusters could internally have some level of dynamics (association and dissociation of monomers), at least it seems to me.
- 2) I even wonder whether some of these larger clusters could not rather reflect a phase separated state. Both (1 and 2) should be addressed in my opinion. An easy analysis would be to quantify the fluorescence of the large clusters in time, though I do not expect to see significant decreases there, even for a dynamic system. (Partial) bleaching and then monitoring recovery should be readily doable. And perhaps the authors can think of additional experiments to probe this aspect of cluster dynamics.
- 3) While I like the primary and secondary nucleation picture, it also raises questions. For instance, what triggers the second? Perhaps size of the primary nucleated cluster, but there does not seem to be a specific lateral size in the data: There are large clusters (even beyond the size of the diffraction limit) following the first mode. For primary nucleation (Fig. 8b), do the authors imagine the new monomer to bind the DNA first – as is drawn and described? Would it not make more sense, in Fig. 8b, for the new monomers to first interact (possibly dynamically) via the PYD domain, which increases the local concentration, and hence next enables DNA binding, if diffusion on the DNA is indeed not occurring? Can the authors see short spikes of fluorescence that may evidence binding (PYD-PYD)? The authors describe the long clusters perpendicular to the DNA as secondary nucleation, which might make sense for a second row of AIM2 proteins as drawn in Fig. 8c, but not for such large structures. What would happen if the PYD-AIM2 proteins would be mixed with PYD monomers? Would this promote secondary nucleation, as the latter is based on PYD while primary needs both?

Other points:

- 1) The main text gives sometimes too much detail (EG line 118-148), while not describing key things like which fluorophore is used and how labelling is performed.
- 2) The 'asymmetric' on line 109 is difficult to understand / picture without more information here.
- 3) The concentration is clearly described in the main text and figure for 4e and f, but not for the reference 4d.
- 4) The argumentation starting on line 257 appears quite weak to me, as the estimations of the KD values are very approximate (also given the difficulty of measuring K_{on}), and the argumentation is based on this number being closer to data of one construct compared to another, determined with a very different technique. This correspondence could be based in part on chance, and hence should not be used to draw conclusions from (line 259-263).
- 5) How often were the behaviors in Figure 5 observed?
- 6) How often were the behaviors in Figure 6 observed? This question is of direct importance in judging the primary-secondary nucleation model that is central to this paper.
- 7) To me, the 3D images in Fig. 6d are more confusing than helpful. In particular on the left one does not see much. Maybe two overlapping 'contour-line' plots as used in maps is more insightful.
- 8) Why would the growth rate be higher for AIM2-AIM2 than for AIM2-DNA growth?
- 9) 'permanence time' does not seem to be an appropriate term.
- 10) The conclusion drawn in line 383-384 seems tenuous.

We are grateful to the reviewers for their insightful comments and thorough critiques of our work. The revised manuscript includes new experiments, results, explanations, and corrections that hopefully have strengthened its quality and addressed the reviewers' concerns. Please, find below a point-by-point response.

Response to the reviewers' comments:

Reviewer #1:

We would like to thank the reviewer for their positive comments and constructive concerns.

“Major issues:

1) The kymographs in the manuscript do not support the growing of DNA-bound AIM2 oligomers which calls into question the major findings of this manuscript. Figure 2b and statement in the text show that within seconds and at AIM2 concentrations of 0.2-1.4 nM, fluorescent AIM2 clusters of various sizes appeared on DNA. However, the kymographs in figures 4b and 7b do not show any increase of fluorescence intensities for single molecule trace up to a few minutes. In fact, the fluorescence intensity actually decreased or disappeared within that time frame perhaps from AIM2 dissociation or photobleaching.”

We thank the reviewer for this suggestion. Oligomer growth in kymographs is more difficult to observe because the scanning laser only monitors binding events happening on the dsDNA molecule, unlike 2D scans that monitor overall oligomer growth. In addition, photobleaching affects the fluorescence intensity in kymographs.

*However, careful analysis of traces indicates fluorescence intensity increase. To facilitate observing this result, we have created heat maps based on changes in the fluorescence intensity corresponding to several kymographs as examples. We have added this information in **Figure 5** of the revised version. These newly added data confirm the results previously shown. The text has been modified accordingly in lines 341 - 360.*

“2) This inconsistency coupled with concern about the potential aggregation of the AIM2 sample prior to DNA binding (see below), suggest that maybe some of the clusters shown in figure 2b were pre-formed before DNA binding, rather than the AIM2 clusters growing on DNA.

Supple figure 2c shows AIM2 cluster/aggregates in the absence of DNA, which is concerning, especially since supplementary figure 3A does not show chromatogram earlier than 30 minute elution time that may reveal larger or aggregated AIM2 samples.”

We have faced several important challenges in this work due to the strong tendency of AIM2 to polymerize in the presence and absence of dsDNA, as this is one of the protein's main functions. Nonetheless, we have been working at concentration values significantly lower than those previously reported to keep AIM2 in monomeric state¹ and use Triton X-100 and glycerol in our buffers to prevent protein oligomerization.

In the new version of the manuscript, we show a full chromatogram for the purification of AIM2-peptide-Alexa 488 with elution time starting at zero minutes (Supplementary Figure 8a). No other signal is observed prior to the AIM2 monomeric signal. It is important to emphasize that we are working at low nanomolar concentration, thus pushing the detection limit of absorbance spectroscopy (micromolar range) in the HPLC system. We multiply the detector signal by a factor of 500 to be able to observe the single peak, which explains the noise and instability of the baseline. We are not showing the original chromatogram because the baseline was instable. However, we have several other representative chromatograms with slightly more stable baselines, but less clear peak, as shown in new Supplementary Figure 8a.

We collected the only peak observed in the chromatogram to perform single molecule studies. This peak corresponds to monomeric AIM2 (Supplementary Figure 8a, b). Despite this, in some instances we have observed oligomers bound to the beads and to DNA (Supplementary Figure 7). We believe oligomerization is minimal as we have not been able to observe AIM2 filaments by TEM, which are prevalent when AIM2 oligomerizes, as previously reported¹. In addition, previous studies in bulk using fluorescence anisotropy indicate that AIM2 is monomeric at concentrations lower than 500 nM¹. Based on previously reported data, we conclude that AIM2 is mainly monomeric at concentrations ~ 0.2 - 14 nM and that the use of single molecule techniques allows to observe few oligomeric species.

*Our results on molecular counting are in accord with previous studies indicating a preferred oligomer size of ~ 20 - 25 AIM2 molecules^{1,2}, thus supporting that our data are derived from the monomeric behavior of the protein and validating the results in **Figures 1 and 2** of the revised version.*

*Importantly, results on k_{off} , oligomer growth rates (the latter determined in many instances starting from 1 or 2 molecules, **Figures 4, 5 and Supplementary Figure 10**), and lack of diffusion of AIM2 along the DNA are derived from single AIM2 molecules based on the photon count analysis that has been corroborated by additional experiments using a single fluorophore attached to dsDNA (please see below). In addition, the different growth mechanisms are proposed based on the observed growth in real time of AIM2 oligomers starting with only few molecules (**Figure 6 and Supplementary Figure 10**).*

“3) A small fraction of aggregated or sticky AIM2 samples that may non-specifically associate with beads could skew the interpretation of AIM2-AIM2 interaction in cluster assembly. The flow cells or buffers may need to include BSA to reduce non-specific interactions and lend confidence to the observed data.”

Following the reviewer's suggestion, all new data (except experiments to test force dependence) have been acquired after passivating the protein channel of the microfluidics flow cell with BSA to further reduce potential non-specific binding, and BSA has been added to all buffers. The newly acquired data in the presence of passivation (**Figure 5 and Supplementary Figure 10**) do not indicate a change in protein behavior compared to data obtained in the absence of BSA. For instance, we have performed a new analysis of cluster growth reporting analogous growth rates ranging from 11.4 s^{-1} to 25.3 s^{-1} (**Supplementary Figure 10**). The mechanisms of cluster growth by DNA binding and AIM2-AIM2 binding are also observed in the new analysis done with passivation (**Supplementary Figure 10**).

We have included the passivation protocol and indicated the new data obtained under these conditions in the Methods section.

“Minor issues:

Figure 1 shows signaling pathways and multiple published structures. Those are not part of the current work and may be more appropriate as a supplementary figure or for a review manuscript.”

Figure 1 has been removed and is now shown as **Supplementary Figure 1**.

“Figure 2c shows that the dominant AIM2 clusters are in $0.1\text{-}1\mu\text{m}^2$ range, which may span a few thousand bp of B form dsDNA. This seems inconsistent with the estimate of 25 AIM2 molecules per cluster given the size of the dsDNA that may bind hundreds of AIM2 molecules.”

We would like to thank the reviewer for this pertinent observation. The apparent discrepancy is related to the resolution of the confocal microscope.

According to the calibration of the confocal microscope using a fluorescent bead of 100 nm in diameter, the point spread function (PSF) measured as full width at half maximum (FWHM) is 351 (X) - 312 (Y) nm using blue excitation laser (**Supplementary Figure 2**). The width at the base of the PSF after Gaussian fit considering the range $\bar{x} \pm 1.5*\sigma$ is 570 nm, and considering the range $\bar{x} \pm 2*\sigma$ is 760 nm. These results indicate that a 100 nm fluorescent bead will appear as a $\sim 600\text{-}700$ nm spot.

We have made an analogous analysis with beads of 23 nm in diameter (**Supplementary Figure 2**). The FWHM (X) and (Y) are 342 nm and 322 nm, respectively, which are very similar to the data obtained for the 100 nm bead, thus indicating the resolution of the confocal microscope. The width at the base of the PSF considering the range $\bar{x} \pm 1.5*\sigma$ is 420 nm, and considering the range $\bar{x} \pm 2*\sigma$ is 560 nm.

The X-ray structure of the AIM2-DNA complex shows a 20-mer dsDNA molecule hosting 2 HIN domains (DNA-binding domain) of AIM2². Based on these data, oligomers of 25 AIM2 molecules (25 HIN domains) would occupy 250 bp, thus spanning 85 nm (0.34 nm per base pair). Therefore, we would expect a FWHM close to 300 nm for the 85 nm cluster of AIM2 molecules, and close to 600 nm in the base, in analogy to the 100 nm fluorescent bead.

To determine the area in the cluster size analysis, we selected all bright pixels with Image J; thus, the base of the PSF, which is significantly wider than the FWHM. The most frequent cluster size is $0.3 \mu\text{m}^2$ - $0.5 \mu\text{m}^2$ equivalent to a diameter of 600 - 800 nm. This number is close to the base of the PSF observed for the 100 nm fluorescent bead used to determine the microscope's resolution, which could correspond to 85 nm (250 bp) hosting 25 molecules or less.

Previously reported TEM studies showed that AIM2 polymerizes into filaments that branch into other filaments¹. Thus, oligomers composed of several 85 nm filaments (with ~ 25 AIM2 molecules each) will result in significantly larger sizes than expected due to the combination of the PSFs of the different filaments.

We would like to illustrate this effect by comparing the cluster at the bottom of Figure 6c (area = $0.88 \mu\text{m}^2$, corresponding to 44 molecules) and the cluster shown in **Supplementary Figure 3a** with 2454 photon counts corresponding to ~ 267 molecules and an area of $1.07 \mu\text{m}^2$ (diameter = $1.16 \mu\text{m}$). The molecular counting method is the same, but the AIM2 filaments are packed differently. The cluster in **Figure 6c** looks less compact than the cluster in **Supplementary Figure 3**. This different filament packing results in clusters of similar area with significantly different number of molecules. The dimensions of the compact cluster in **Supplementary Figure 3** (diameter = $1.16 \mu\text{m}$) and the number of molecules (~ 267) match based on the dimensions of AIM2, ~ 5 nm wide and ~ 8 nm long, as it is composed of two domains connected by a linker. This result validates our molecular counting method and confirms that the dimensions of the clusters do not accurately represent the actual number of protomers in the polymers due to differential packing and the optical resolution.

This new information and explanation have been included in the revised version of the manuscript (**Supplementary Figure 2**) showing the 2D images of 100 and 23 nm beads and the analysis of the PSF, and in lines 104 - 125.

To provide additional information on the optical resolution of the confocal microscope, we have obtained confocal images of a single fluorophore, Atto 488, attached to biotinylated λ -DNA (gift from Prof. Muñoz, UC Merced, sample prepared by Lumicks). Atto 488 and Alexa 488 have almost identical molecular structures^{3,4} (**Supplementary Figure 6d**) and similar quantum yield (QY); $QY_{\text{Alexa 488}} = 0.92^5$; $QY_{\text{Atto 488}} = 0.80^6$. The figure below (Figure Rev_1.1) shows the population and size distribution of 22 images of Atto 488 on the DNA, analyzed with ImageJ with the same tools used for the analysis of AIM2 cluster sizes. The average size of the fluorescent signal resulting from a single fluorophore is $0.15 \pm 0.08 \mu\text{m}^2$ ($\text{Ø} = 437 \text{ nm}$). A significantly larger number of Atto 488 molecules would be expected in a fluorescent spot with a diameter of 437 nm.

Figure Rev_1.1. *a* Representative two-dimensional scan of Atto 488 (white circle) attached to λ -DNA (scale bar is 3 μm). *b* Size distribution of Atto 488 fluorescent signal from images acquired by excitation at $\lambda = 488 \text{ nm}$ ($N = 22$).

“Figures 5-6 show that the cluster size grew by 3-4 fold within a few seconds, which is inconsistent with the 13-15-molecule size of oligomers shown in figure 3 if the growth starts with 1-2 AIM2 molecules. Suggest increase the time frame to see if the cluster size may increase to the estimated dominant species of 13-15 oligomers.”

Our data indicate that oligomers up to 25 molecules are dominant (**Figure 2a**). Within this size range, oligomers of 13 - 15 molecules are only slightly more populated than for example the 4 - 6 and 10 - 12 sizes. Figures 5 and 6 in the original version are Figures 4 and 6 in the revised version. Figure 4 shows clusters growing to 4, 8 and 10 molecules. New version of **Figure 6** shows clusters growing to 10, 14 and 24 molecules. All these cluster sizes are also very populated as shown in **Figure 2b**. Following the reviewer’s suggestion, we have performed new cluster growth analyses, which show examples of size increase by ~ 3 -fold in **Figure 5** (top kymograph) and **Supplementary Figure 10** (panels *a* and *c*)

“Figure 6d shows $t=1\text{s}$ and $t=2\text{s}$ on the left, whereas $t=1\text{s}$ and $t=4\text{s}$ on the right. Such comparison may not accurately represent how AIM2-DNA binding compare with AIM2-AIM2 binding. For consistency, the authors should show $t=4\text{s}$ instead of $t=2\text{s}$ on the left.”

The fluorescence intensity at $t = 4 \text{ s}$ for oligomer growth via AIM2-DNA binding decreases because of photobleaching, thus not properly representing growth. This effect is also shown in **Figure 4**, (indicated in the original version of the manuscript), where the fluorescence intensity decreases at certain time points but shows an overall upward trend. Nonetheless, the rates are reported as net increase in fluorescence intensity per second, which removes the dependence with time, thus making the comparison adequate to illustrate the different growth mechanisms by comparing the surface plots in **Figure 6**.

To further support these data, we have analyzed additional clusters to show oligomer growth by AIM2-DNA binding and AIM2-AIM2 binding, as well as the corresponding growth rates. These new results are shown in **Supplementary Figure 10**.

“Supple figure 1a shows 2-step photodepletion followed by 2 1-step photodepletion for a total of 4-step. However, because the photon counts are so scatted, a total of 3 1-step photodepletion seems to fit the data just as well as the 2-step+1-step+1step photodepletion.

This will change the estimate of the average number of photons per single fluorophore in supple figure 1b, and the estimate of the number of AIM2 in each cluster. Suggest using photobleaching method to estimate single AIM2 photon count.”

We have followed two approaches that confirm our original analysis of the number of photon counts per fluorophore.

1) *Automated detection of steps:*

*The trajectories of fluorescence intensity decay vs. time of AIM2 oligomers exposed to laser excitation display complex behavior due to the large number of fluorophores present in the oligomers (**Supplementary Figures 4 and 5**). As it is common in the presence of complex behavior, our analysis of photobleaching steps was done by visual inspection because an automated method could detect non-existing states.*

*Nonetheless, for the less complex trajectories we have used a recently published method (“Autostepfinder”⁷) to automatically find steps in trajectories. This method finds the same number of steps we detected by visual inspection for several representative trajectories. Specifically, for the trajectory shown in **Supplementary figure 4a**, the “Autostepfinder” method can detect 2 or 3 steps with identical quality of fit as shown in **Supplementary Figure 5**. Our visual inspection indicates 3 steps and we have decided to trust this result. For the automated method to work, we had to filter data by removing all points with values larger than the average \pm standard deviation calculated in intervals of 10 s. In another representative example, the “Autostepfinder” method detects 2 steps in the raw data, matching what we originally determined by visual inspection (**Supplementary Figure 5**). In addition, the sizes of the steps are also in agreement with what we determined originally (**Supplementary Figure 5**).*

2) *Photon counts emitted by a single fluorophore using a different molecule:*

*We do understand that the visual inspection of fluorescence trajectories can be prone to certain biases. To increase confidence in our data, we have determined emitted photon counts of the single fluorophore (Atto 488) attached to biotinylated λ -dsDNA using exactly the same experimental method used for AIM2. Our results (**Supplementary Figure 6**) show that the single Atto 488 bound to dsDNA emits 12 ± 2 photons. This value is very close to the 11 photon counts determined by the step analysis. It is reasonable to consider that Alexa 488 and Atto 488 emit a similar number of photons as both fluorophores have very similar molecular structures (**Supplementary Figure 6**) and similar quantum yields, as indicated above.*

*This new information has been included in the revised version of the manuscript in **Supplementary Figures 4, 5 and 6**. The corresponding explanations have been added in the text in lines 188 - 197.*

“Since optical traps were used, it should be clarified whether the measurements on AIM2-DNA association observed here are dependent on the amount of force applied or not.”

*Following the reviewer’s suggestion, we have investigated whether dsDNA stretching affects AIM2 binding. All experiments shown in the manuscript were acquired at 17 pN, which stretches the dsDNA to 16 μm under our experimental conditions. We have increased the force to 40 pN leading to a dsDNA end-to-end distance of 16.5 μm . The dwell time analysis of a total of 172 single molecule traces in kymographs acquired under these conditions results in a k_{off} of $0.52 \pm 0.08 \text{ s}^{-1}$ (**Supplementary Figure 9**). This result indicates that the additional force reduces the time AIM2 remains bound to dsDNA. It is possible that the additional force causes potential distortions of the dsDNA structure, thus affecting AIM2 binding. Stretching the dsDNA with small forces (20 pN) leads to a decrease in the inter-strand distance, resulting in dsDNA overwinding as shown by studies using all-atom molecular dynamics simulations⁸. In addition, it has been demonstrated that dsDNA stretching affects protein function. As an example, dsDNA stretching has been shown to induce off-target activity of the endonuclease Cas9 due to structural disruptions in the dsDNA molecule⁹.*

*This information is included in the revised version of the manuscript: **Supplementary Figure 9** and lines 262 - 275 in the text.*

“Scale bars are missing for many of the figures.”

We apologize for this oversight. All scale bars have been included in the revised version of the manuscript.

Reviewer #2:

We would like to thank the reviewer for the expressed interest in this work and the very pertinent concerns.

1. “- Most importantly, I wonder about the counting method the authors use. My main concern actually arose after observing the growth experiments of Figure 6. In particular, Figure 6c shows an elongated cluster that from the analysis seems to be significantly larger than 0.5 micron. Given the size of a 324-residue protein like AIM2 (perhaps 2-3 nm), such a cluster should contain hundreds of protein molecules. Yet the authors conclude that it contains only ~20 molecules! The counting method should therefore be revised and also explained better.”

We would like to thank the reviewer for this very important observation. There are several factors playing a role in the apparent discrepancy.

*The cluster with 20 molecules (23 molecules after corrections, please see responses to points #4 and #16) does not correspond to the vertical one but the cluster of the first 2D scan at the top of **Figure 6c**. The diameter of this cluster is 781 nm.*

The apparent discrepancy between the dimension of the cluster and the number of molecules is related to the resolution of the confocal microscope. We have obtained new data to show that a fluorescent cluster of ~ 600 - 700 nm in diameter can easily correspond to a ~ 100 nm fragment of dsDNA with 25 AIM2 molecules attached to it.

*According to the calibration of the confocal microscope using a fluorescent bead of 100 nm in diameter, the point spread function (PSF) measured as full width at half maximum (FWHM) is 351 (X) - 312 (Y) nm using blue excitation laser (**Supplementary Figure 2**). The width at the base of the PSF after Gaussian fit considering the range $\bar{x} \pm 1.5 \cdot \sigma$ is 570 nm, and considering the range $\bar{x} \pm 2 \cdot \sigma$ is 760 nm. These results indicate that a 100 nm fluorescent bead will appear as a ~ 600 -700 nm spot.*

*We have made an analogous analysis with beads of 23 nm in diameter (**Supplementary Figure 2**). The FWHM (X) and (Y) are 342 nm and 322 nm, respectively, which are very similar to the data obtained for the 100 nm bead, thus indicating the real resolution of the confocal microscope. The width at the base of the PSF considering the range $\bar{x} \pm 1.5 \cdot \sigma$ is 420 nm, and considering the range $\bar{x} \pm 2 \cdot \sigma$ is 560 nm.*

The X-ray structure of the AIM2-DNA complex shows a 20-mer dsDNA molecule hosting 2 HIN domains (DNA binding domain) of AIM2². Based on these data, the oligomers of 25 AIM2 molecules (25 HIN domains) we observe would occupy 250 bp, thus spanning 85 nm (0.34 nm per base pair). Therefore, we would expect a FWHM close to 300 nm for the 85 nm cluster of AIM2 molecules, and close to 600 -700 nm in the base, in analogy to the 100 nm fluorescent bead.

To determine the area in the cluster size analysis, we selected all bright pixels with Image J; thus, the base of the PSF, which is significantly wider than the FWHM. The most frequent cluster size is $0.3 \mu\text{m}^2$ - $0.5 \mu\text{m}^2$ equivalent to a diameter of 600 - 800 nm. This number is close to the base of the PSF observed for the 100 nm fluorescent bead used to determine the microscope's resolution, which could correspond to 85 nm (250 bp) hosting 25 molecules or less.

Previously reported TEM studies showed that AIM2 polymerizes into filaments that branch into other filaments¹. Thus, oligomers composed of several 85 nm filaments (with ~ 25 AIM2 molecules each) will result in significantly larger sizes than expected due to the combination of the PSFs of the different filaments.

We would like to illustrate this effect by comparing the cluster at the bottom of Figure 6c (area = $0.88 \mu\text{m}^2$, corresponding to 44 molecules) and the cluster shown in Supplementary Figure 3a with 2454 photon counts corresponding to ~ 267 molecules and an area of $1.07 \mu\text{m}^2$ (diameter = $1.16 \mu\text{m}$). The counting method is the same, but the AIM2 filaments are packed differently; the cluster in Figure 6c looks less compact than the cluster in Supplementary Figure 3. This different filament packing results in clusters of similar area with significantly different number of molecules. The dimensions of the cluster in Supplementary Figure 3 (diameter = $1.16 \mu\text{m}$) and the number of molecules match based on the dimensions of AIM2, ~ 5 nm wide and ~ 8 nm long, as it is composed of two domains connected by a linker. This result validates our molecular counting method and confirms that the dimensions of the clusters do not accurately represent the actual number of protomers in the polymers due to differential packing and the optical resolution.

This new information and explanation have been included in the revised version of the manuscript (Supplementary Figure 2) showing the 2D images of 100 and 23 nm beads and the PSF analysis, and in lines 104 - 125.

We hope to have explained the counting method better in the revised version (lines 185 - 197 and Methods section). We have included the use of automated analysis of the steps in fluorescence intensity decays (Supplementary Figure 5. Please see additional explanation below for point # 2) and we have determined the photon counts of a single fluorophore attached to dsDNA (Supplementary Figure 6. Please see additional explanation below for point #8) to further support our molecular counting method.

To provide additional information on the optical resolution of the confocal microscope, we have obtained confocal images of a single fluorophore Atto 488 attached to biotinylated λ -DNA (gift from Prof. Muñoz, UC Merced, sample prepared by Lumicks). Atto 488 and Alexa 488 have almost identical molecular structures (Supplementary Figure 6)^{3,4} and similar quantum yield (QY); $QY_{\text{Alexa 488}} = 0.92^5$; $QY_{\text{Atto 488}} = 0.80^6$.

Figure Rev_2.1 below shows the population and size distribution of 22 images of Atto 488 on the DNA, analyzed with ImageJ with the same tools used for the analysis of AIM2 cluster sizes. The average size of the fluorescent signal resulting from a single fluorophore is $0.15 \pm 0.08 \mu\text{m}^2$ ($\emptyset = 436$ nm). A significantly larger number of Atto 488 molecules would be expected in a fluorescent spot with a diameter of 436 nm.

Figure Rev_2.1. a Representative two-dimensional scan of Atto 488 (white circle) attached to λ -DNA (scale bar = 3 μm). **b** Size distribution of Atto 488 fluorescent signal from images acquired by excitation at $\lambda = 488 \text{ nm}$ ($N = 22$).

2. - “In relation to this, the assignment of steps in sup. figure 1 is not so obvious- was any analytical method used for this assignment? How was it determined that some steps are due to two molecules, for example? This is crucial for the counting analysis.”

We apologize for not having explained this important aspect of the data analysis with sufficient clarity.

*Our data from kymograph analysis consistently indicate that oligomers formed by more than 3 AIM2 molecules do not dissociate from the dsDNA as the fluorescence intensity lasts as long as the kinetic experiment, oftentimes even more than 20 minutes (please see manuscript line 255, **Figure 3b and Supplementary Table 1**). Therefore, we have concluded that the decay in fluorescence intensity for AIM2 oligomers is not dissociation but photobleaching.*

We have followed two approaches that confirm our original analysis of the number of photon counts per fluorophore.

1) *Automated detection of steps:*

*The fluorescence intensity decays of AIM2 oligomers typically display complex behavior due to the large number of fluorophores present. As it is common in the presence of complex behavior, our analysis of photobleaching steps was done by visual inspection⁷ because an automated method could detect non-existing states. The number of photon counts corresponding to the steps found in the visual inspection ranges from 2 to 24 (**Supplementary Figure 4b**). The corresponding histogram was fitted to a Gaussian distribution that is centered at 11 photon counts. Thus, it was assumed that this number of photon counts corresponds to the bleaching of one fluorophore, which should be statistically predominant compared to the simultaneous bleaching of 2 or more fluorophores.*

*In the revised version, we have used a recently published method (“Autostepfinder”⁷) to find steps automatically. This method finds the same number of steps we detected by visual inspection for some representative trajectories. Specifically, for the trajectory shown in **Supplementary Figure 4a**, the “Autostepfinder” method can detect 2 or 3 steps with identical quality of fit as shown in **Supplementary Figure 5**. Our visual inspection indicates 3 steps and we have decided to trust this result. For the automated method to work, we had to filter data by removing all points with values larger than the average \pm standard deviation calculated in intervals of 10 s. In another representative example, the “Autostepfinder” method detects 2 steps in the raw data, matching what we originally determined by visual inspection (**Supplementary Figure 5**). In addition, the sizes of the steps are also in agreement with what we determined originally (**Supplementary Figure 5**).*

2) *Photon counts emitted by a single fluorophore using a different molecule:*

*We do understand that the visual inspection of fluorescence trajectories can be prone to certain biases. To increase confidence in our data, we have determined emitted photon counts of the single fluorophore (Atto 488) attached to biotinylated λ -dsDNA using exactly the same experimental method used for AIM2. Our results (**Supplementary Figure 6**) show that the single Atto 488 bound to dsDNA emits 12 ± 2 photons. This value is very close to the 11 photon counts determined by the step analysis. It is reasonable to consider that Alexa 488 and Atto 488 emit a similar number of photons as both fluorophores have very similar molecular structures (**Supplementary Figure 6**) and similar quantum yields, as indicated above.*

*This new information has been included in the revised version of the manuscript in **Supplementary Figures 4, 5 and 6**. The corresponding explanations have been added in the text in lines 188 - 197.*

3. - “As another related point, surely the assignment of 11 photons for the signal of a single molecule is just approximate. This means that there is an error that propagates into each determination of number of molecules- this error should be estimated and used for each calculation.”

We would like to thank the reviewer for this important observation. The error in the estimation of the photon counts per fluorophore using the step analysis is 36% (11 ± 4) and the error using a different single fluorophore molecule is 17% (12 ± 2). We have used the smallest error for error propagation measurements. After considering error propagation:

$$\frac{\Delta S}{S} = \frac{\text{error in PCs for single fluorophore}}{\text{PCs single fluorophore}}$$

$$\Delta S = S \cdot 0.17$$

Where, ΔS is the error and S is the size of the cluster (number of molecules).

*This information is now included in the legend of **Figure 2** of the revised version.*

4. - “In this respect, one wonders whether fluorescence is quenched in clusters, leading to under-counting. There is some discussion of this point in the paper, but I did not see a conclusive test of this possibility.”

We have followed the reviewer’s recommendation. Previous work in this regard shows evidence that fluorophore clustering could result in underestimation of the number of fluorophores based on measurements of total fluorescence^{10,11}. In fact, homo-FRET¹² and Stokes shift processes¹³ resulting from fluorophore clustering have been observed.

*Our confocal microscope has two detection channels in blue and red with two bandpass filters: 512/25 nm and 700/100 nm. The fluorophore we use to label AIM2 is Alexa 488, with maximum emission at 520 nm. We have observed emission at red wavelengths for clusters emitting more than 100 blue photon counts (PCs). This emission in the red is fairly constant with an average of 14 ± 4 PCs for clusters from 100 up to ~ 575 blue PCs and increases for larger clusters. This analysis is now included in **Supplementary Figure 3**. We have ruled out that the emission in the red is caused by bleed-through because it does not increase linearly with the increase in the emission of blue photon counts.*

To consider the loss in blue photons due to fluorophore emission at red wavelengths, we have estimated that the energy of a blue photon is 1.4 times higher than that of a red photon using the ratio of the wavelengths at the center of the filter bandwidth; 700/512. Thus, the average of 14 red photons is equivalent to 10 blue photons. This ratio was used to correct for the underestimation of fluorescence at blue wavelengths due to the emission in red in the range from 100 to 575 blue photons. This means that for clusters emitting from 100 to 575 blue photon counts, one fluorophore is added to the original count.

This new information is explained in the text in lines 155 - 176, Supplementary Figure 3, and in the legend of Figure 2. All figures/tables showing molecular counting have been corrected by adding one fluorophore due to the emission at red wavelengths (from clusters emitting 100 to 575 blue photon counts) and for the labeling efficiency that was determined more accurately with mass spectrometry (please see point #16 below).

We would like to emphasize that the analysis explained above for clusters in Figure 6c and Supplementary Figure 3 (please see response to point #1), validates our molecular counting method and points to filament packing as an important factor determining cluster dimensions.

5. - “The disappearance of fluorescent signals, as well as steps in signals, can be attributed to either photobleaching or dissociation of proteins. I am not sure I see a clear way to distinguish between the two in the paper.”

Our data from the kymograph analysis consistently indicate that oligomers formed by ≥ 3 AIM2 molecules do not dissociate from the dsDNA as the fluorescence intensity lasts as long as the kinetic experiment, oftentimes even more than 20 minutes (please see manuscript line 255,

Figure 3b, and Supplementary Table 1). Therefore, we have concluded that the decay in fluorescence intensity for AIM2 oligomers is not dissociation but photobleaching.

6. - “The very large clusters reported in Supp Figure 2 are not well characterize and it is not clear when they appear and how their appearance might affect the results of the work.”

*We have not characterized the large clusters. Ours and previously reported data¹, suggest that these are minor species resulting from slight oligomerization of AIM2, despite the many approaches we tried to prevent their formation. For instance, we start with monomeric protein as the size exclusion chromatogram shown in **Supplementary Figure 8** indicates that AIM2 is monomeric at nanomolar concentrations right before starting the single molecule experiments. We add Triton X-100 and glycerol in all our buffers to prevent protein oligomerization.*

Importantly, these large clusters do not represent the main population because AIM2 massively forms filaments upon oligomerization¹. However, we have not been able to observe AIM2 filaments by TEM, which are prevalent as previously reported by other groups¹. In addition, previous studies in bulk using fluorescence anisotropy indicate that AIM2 is monomeric at concentrations lower than 500 nM¹. Based on our TEM data and previously reported data, we can conclude that AIM2 is mainly monomeric at concentrations ~ 1 -14 nM and that the use of single molecule techniques allows to observe the few oligomeric species.

However, oligomerization prevented us from accurately determining the effective concentration of monomers, thus affecting the measurement of k_{on} . A range of values is proposed for k_{on} and K_D due to this uncertainty. This information is indicated in the manuscript in lines 297 - 304.

Based on the monomeric state of the protein immediately before the start of the single molecule experiments, our data on cluster size and molecular counting mainly reflect monomeric behavior. With respect to molecular counting, it is important to mention that our results are in accord with previous studies indicating a preferred oligomer size of 20-24 AIM2 molecules^{1,2}, thus supporting that our data are derived from the monomeric behavior of the protein.

*Importantly, all results on k_{off} and oligomer growth rate (in many instances starting from 1 or 2 molecules, **Figures 4, 5, 6 and Supplementary Figure 10** in the revised version), and lack of diffusion of AIM2 along the DNA are derived from single AIM2 molecules based on the photon count analysis that has been corroborated by additional experiments using single fluorophore attached to dsDNA (please see response to point #8). In addition, the different growth mechanisms are proposed based on the observed growth in real time of AIM2 oligomers starting with only few molecules (**Figure 6 and Supplementary Figure 10**).*

“Additional point by order of appearance in the paper:”

7. - “It is not clear whether the cluster size distribution is fixed or time-dependent. Further, it seems from Figure 2C that small clusters might be missed by the analysis.”

The cluster size distribution is derived from the analysis of 2D fluorescence scans acquired right after placing the trapped beads on the protein channel. Therefore, the distribution is fixed. These

2D scans indicate that AIM2 oligomers can grow to different size distributions in many instances in less than 2 s. The analysis shows that not all sizes are equally populated at a fixed time. To investigate oligomer formation, we have acquired movies that have allowed us to observe oligomer growth in seconds and these data are time dependent (**Figures 4 and 6**). We have performed new cluster growth analyses shown in **Figures 5 and Supplementary Figure 10** to provide additional examples of time dependent growth corroborating our original findings.

We did a thorough analysis on cluster size that we believe represents the behavior of AIM2.

8. - “The specific detectors used in the experiment should be mentioned and how the authors have arrived at a counting underestimation of 6% should be explained. Photon counting vs. molecular number can in principle be tested and calibrated independently with other molecules.”

A) We use a commercial C-Trap (B08 model) instrument from the company Lumicks. We have asked about the specific detector type but unfortunately, manufacturer’s name and detector model are proprietary information. Nonetheless, Lumicks has shared with us the information about the detectors in our C-Trap shown below:

- Detector dead time → 35 ns
- Photon detection efficiency (PDE) → at 650 nm, 75%; at 830 nm 50%, at 512 nm, 50 – 70%

B) To estimate photon count underestimation due to detector dead time, we have followed an idealized dead time model¹⁴ that assumes the detector is not affected by events happening during the dead time, τ ($\tau = 35$ ns). This implies that the detector is dead for a fixed time after each event¹⁴.

For a counting rate ‘ m ,’ the fraction of time during which the detector is dead is ‘ $m \cdot \tau$.’ The fraction of time during which the detector is detecting is ‘ $1 - m \cdot \tau$.’ The number of true events per unit of time is ‘ n .’

Thus, the number of true events that can be detected is:

$$n = \frac{m}{1 - m \cdot \tau}$$

We have used a pixel dwell time of 50 μ s in which we detect 11 photons per fluorophore (according to our analysis shown in Supplementary Figures 4, 5 and 6), which gives a detection rate of $220 \cdot 10^3$ photons/s.

In this case, the real photon count is:

$$n = \frac{220 \cdot 10^3}{1 - 220 \cdot 10^3 \cdot 35 \cdot 10^{-9}} = 221,707 \text{ photons/s}$$

Thus, there is an underestimation of 0.8% for one fluorophore.

If the cluster has 10 fluorophores, the detection rate will be $2200 \cdot 10^3$ photons/s. However, the real number of photons/s will be:

$$n = \frac{2200 \cdot 10^3}{1 - 2200 \cdot 10^3 \cdot 35 \cdot 10^{-9}} = 2,383,532 \text{ photons/s}$$

Thus, resulting in an underestimation of 7.7% for a cluster of 10 fluorophores. This percentage results in approximately one fluorophore not being counted in a cluster of 10.

We originally did this calculation with 10 photon counts per fluorophore, instead of 11 just to round the number, and this is the reason why the original underestimation was stated as 6%. In the revised version we have used the actual 11 photon counts leading to an underestimation of 7.7% (~ 8%).

This information is now explained in the Methods section and the text has been changed accordingly (lines 135 - 141).

*We have performed photon counting with a different molecule as suggested by the reviewer using a single fluorophore attached to λ -DNA (please see response to points #1 and #2 above). These results are explained in lines 188 - 197, shown in **Supplementary Figure 6** and corroborate our original results.*

9. - “Page 5, top paragraph- it is not clear what sigmoidal response is being discussed here and why.”

It has been reported previously that the binding of AIM2 to DNA is an all-or-none process¹ typically resulting in sigmoidal binding curves. This information is not important and has been removed for clarity. The message we aim to convey is that the single molecule data presented in the manuscript, showing clusters of different sizes and shapes, provide additional information on the binding of AIM2 and dsDNA, previously described as an all-or-none process. We have changed the text accordingly (lines 204 - 207).

10. - “Terminology in the third paragraph of the same page: the authors do not perform any ‘correlative’ microscopy here; it is not clear what ‘genomic position’ means.”

What we meant to say by using the word “correlative” is that the equipment used in this work combines optical traps and confocal fluorescence microscopy. We have removed the term to avoid confusion since Figure 1 in the revised version describes the technique used.

By “genomic position” we meant the position on the dsDNA molecule. The kymographs report the change in fluorescence intensity as a function of time and position on the dsDNA. For clarity, we have removed this word and now use “position on the dsDNA.”

11. - “Page 6, second paragraph: ‘we can assume that the koff is zero for the longest DNA’- why should it be zero?”

The experiments in bulk use 72 nM protein concentration and report that the AIM2 oligomers are composed of ~ 20 molecules^{1,2}. Based on our data, oligomers of this size stay permanently attached to the dsDNA. Thus, in the limit of permanent attachment, we assume k_{off} is zero to estimate a value of the k_{on} . We have modified the text to clarify this assumption based on our single molecule data (lines 282 - 285).

12. - “It is worrisome that the KD obtained here is so much larger than estimated by others. This could indicate that kon is underestimated significantly.”

We do understand this important concern from the reviewer.

Table 1 lists the K_D values obtained by other groups for the complex between the DNA-binding domain of AIM2 (HIN) and DNA. These values range from 176 to 584 nM. The K_D value for full-length AIM2 obtained by analyzing single molecules ranges from 160 to 780 nM, thus within the same order of magnitude. Because we used single molecule data (only selecting traces corresponding to single molecules), protein-protein oligomerization via PYD is not possible.

It has been demonstrated using fluorescence anisotropy in bulk that protein-protein oligomerization in AIM2 increases the affinity for dsDNA, starting with K_D values in the hundreds of nM (comparable to our data) to < 3 nM¹. This conclusion is derived from the comparison of K_D values determined for the binding of AIM2^{HIN} and AIM2^{FL} to dsDNA¹. These experiments cannot isolate the behavior of single molecules and thus measure a combined effect of protein-DNA binding and protein-protein binding. By using single molecule techniques, we have been able for the first time to determine the true range of K_D for AIM2-DNA binding in the absence of protein-protein interaction, and our values match previously reported data for AIM2-dsDNA binding in the absence of PYD. This agreement supports our data.

Our results on the different k_{off} for a single AIM2 molecule (0.3 s⁻¹) compared to larger oligomers (permanent attachment) reflect the different K_D (k_{off}/k_{on}) in the absence and presence of protein-protein oligomerization.

13. - “Page 7- why are the growth rates not linear with concentration? When the concentration is increased four-fold and then almost ten-fold, the growth rate increases only by a factor of less than 2 and then by a factor of 5, respectively.”

*We thank the reviewer for this observation that has prompted us to analyze the growth of additional clusters. We originally observed a tendency of growth rate increase with concentration for average rate values corresponding to 4 - 5 clusters. Representative examples of the new cluster growth analysis are shown in **Figure 5** (in the form of kymographs) and **Supplementary Figure 10** (2D fluorescence scans). These new data suggest that the starting number of molecules could affect cluster growth rate. Thus, linearity in growth rate and concentration is not expected. In addition, other factors could have an influence on cluster*

growth rate such as the mode of interaction between protomers and filament packing. These results are more clearly explained in the revised version in lines 341 - 360.

14. - “Page 10- ‘AIM2 does not need to diffuse’- proteins diffuse because of their interactions, not because they need or do not need. The question arises as to what makes AIM2 static when on the DNA, even though its interaction with the DNA is not that strong. This is surprising and requires further explanation.”

We have deleted the word “need to” as it is not scientifically appropriate, and we thank the reviewer for pointing this out.

*Our results on the different k_{off} for a single AIM2 molecule (residence time of 3.3 s) compared to larger oligomers (permanent attachment) reflect the different behavior in the absence and presence of protein oligomerization. The expected behavior for AIM2 is to be static on the DNA for sufficiently large clusters (≥ 3 molecules); whereas single AIM2 molecules stay bound to the DNA for an average of 3.3 s, which might not be enough time to observe diffusion. However, we have not observed motion even for the longer residence times of single AIM2 molecules (**Figure 3**).*

We have added this information in the manuscript for further explanation (lines 413 – 424).

*The interactions of proteins with DNA are complex. The strength of the interaction plays an important role, but other factors can influence whether a protein shows diffusive motion on the DNA. For example, the human protein complex XPC-RAD23B recognizes DNA lesions as part of the nucleotide excision repair mechanism¹⁵. Recent studies have shown that XPC-RAD23B binds to λ -DNA in a sequence-independent manner (like AIM2) and shows three distinct and simultaneous types of motion: diffusive, immobile, and constrained (please see **Figure Rev_2.2** below). The population of each type of motion depends on the salt concentration and the presence of AT-rich regions in the λ -DNA. However, the three motion types coexist in all conditions¹⁵. The three types of motions have also been observed in the XPC yeast ortholog Rad4-Rad23¹⁶.*

Figure 2. Three distinct types of motion of XPC-RAD23B on undamaged DNA (A) Kymographs for different types of motion of XPC-RAD23B. From top to bottom, diffusive motion, immobile state, constrained motion and transition between two distinct states are displayed. The white box in the kymographs showing the immobile state and constrained motion represents the zoom-in view of trace. The constrained motion is clearly distinguished from the immobile state. The solid and dotted yellow lines stand for diffusive motion and constrained motion, respectively. The black arrow heads indicate barrier positions. (B) Relative fraction of each motion according to salt concentration. The error bars were obtained by the standard deviation of multinomial distribution. (C) Relative fraction of transitions out of total traces according to salt concentration. The error bars were obtained by the standard deviation of binomial distribution.

Figure Rev_2.2. Figure retrieved from reference 15.

15. - “Details about the microfluidic system are missing- is it home-built? Commercial? What prevents material from diffusing from channel 2 to the other channels? What is the flow rate in the other channels? What is the depth of the device? Is the whole depth being imaged?”

The multi-channel laminar flow microfluidics system is commercial and built by Lumicks. The laminar flow prevents the material to mix or diffuse between the different channels. The flow rate is typically fixed at a pressure of 0.3 bar. Image and kymograph acquisition was done at zero flow rate in the protein channel while other channels were flowing.

The glass of the flow cell under the channel is ~ 175 μm wide and the channel width is 100 μm . We are only taking images in the confocal plane with a depth of ~ 1 μm .

We have added new information related to the flow cell and pressure flow in the Methods section.

16. - “Have the authors assessed the labeling efficiency? This is very important in their

estimates of cluster sizes, as multiple unlabeled proteins can lead to significant underestimated counting.”

We have assessed the labeling efficiency more accurately and show new data to demonstrate our results.

To make sure each AIM2 molecule is labeled with no more than 1 fluorophore we have followed a specific labeling strategy:

1. A short peptide with sequence GGGC was labeled with Alexa 488 via maleimide reaction in the free thiol group of the amino acid cysteine. The labeled peptide was purified, and the labeling efficiency was determined by mass spectrometry. To provide a more accurate value of the labeling efficiency, we have determined the average efficiency of several reactions (N = 4) resulting in an average efficiency of 90%.

2. The labeled peptide is attached covalently to AIM2 via sortase transpeptidation¹⁷. The efficiency of this transpeptidation is close to 100% as determined by mass spectrometry: the labeled protein was observed with matching experimental and theoretical mass and the unlabeled protein was not detected.

*These new data are shown in **Supplementary Figure 8c** and explained with more detail in the **Methods** section.*

All results on number of molecules per cluster have been updated to reflect the more accurate measurement of the labeling efficiency and the loss of fluorescence due to emission in the red from 100 to 575 blue photon counts. After these corrections, the dominant populations have not changed; most clusters are smaller than 25 molecules.

17. - “Page 20- Is it important to show molecular structure of the proteins studied here?”

The figure showing the molecular structures has been removed from the main text and it is now shown in Supplementary Figure 1.

18. - “Page 21- size of scale bars of Fig. 2b is missing. Please check all figures.”

We apologize for this oversight. All figures have the scale bar in the revised version.

Reviewer #3:

“This paper reports on single-molecule measurements of the assembly of AIM2 on dsDNA, a step in an inflammatory response. The study provides dynamic information on this assembly process, adding to existing biochemical and structural work. Overall, I found the aim of the experiments interesting and appropriate for a journal like Nature Communications. However, some of the work is focused on initial characterization, and while they are important, they are more an opening to other considerations that are missing, in my opinion. More specifically:”

We would like to thank the reviewer for their comments and for pointing out important considerations on AIM2 oligomerization.

“1) The large AIM2 assemblies are identified as clusters, and the language generally indicates that the authors think of them as static oligomers. While I agree with the conclusions that the larger clusters remain bound, and hence are static in that sense, the clusters could internally have some level of dynamics (association and dissociation of monomers), at least it seems to me.”

“2) I even wonder whether some of these larger clusters could not rather reflect a phase separated state. Both (1 and 2) should be addressed in my opinion. An easy analysis would be to quantify the fluorescence of the large clusters in time, though I do not expect to see significant decreases there, even for a dynamic system. (Partial) bleaching and then monitoring recovery should be readily doable. And perhaps the authors can think of additional experiments to probe this aspect of cluster dynamics.”

Response to points 1 and 2: We thank the reviewer for sharing these interesting ideas.

1. Studies using TEM have shown that AIM2 oligomerizes forming filaments and filament bundles in the absence and presence of dsDNA, and this process has been considered irreversible^{1,18}. For example, adding a nuclease to the dsDNA-AIM2 complex does not result in filament disassembly once formed¹⁸. These previously reported data do not support the hypothesis that AIM2 oligomers undergo association and dissociation of monomers. However, we were enthusiastic to test the dynamics of the oligomers/filaments as this could inform on the overall dynamics of the inflammasome.

[REDACTED]

[REDACTED]

[REDACTED]

[REDACTED]

Figure Rev_3.3. Excerpt from Figure 7 of reference 22. “Morphology, Stoichiometry, and Pro-IL-1 β Processing in Inflammasomes.(B) Immunogold EM on ultrathin cryosections from ASCFL-eGFP-transfected COS-1 cells. The ASC-containing compact structure is densely decorated by gold particles (10 nm). N, nucleus; NM, nuclear membrane.”

[REDACTED]

Figure Rev_3.4. Retrieved from reference 23.

“3) While I like the primary and secondary nucleation picture, it also raises questions. For instance, what triggers the second? Perhaps size of the primary nucleated cluster, but there does not seem to be a specific lateral size in the data: There are large clusters (even beyond the size of the diffraction limit) following the first mode.

We thank the reviewer for this question and thoughts about the double nucleation mechanism.

We would like to emphasize that the analogy to the primary and secondary nucleation mechanism serves the purpose of explaining the two types of growth we have observed.

In the case of hemoglobin S, the homogeneous (primary) nucleation consists in the formation of a fiber via monomer oligomerization. The heterogeneous (secondary nucleation) occurs by growth of an oligomer (eventually forming a second fiber) on the surface of the initial fiber²⁴.

*We have shown that AIM2 tends to form clusters smaller than 25 molecules. It is possible that beyond this size the probability of growth by PYD-PYD only binding (analogous to secondary nucleation) increases due to a larger surface of exposed PYD domains, although we have observed this type of growth in smaller clusters as shown in **Figure 6**.*

On another note, if clusters grew only via AIM2-DNA binding (analogous to primary nucleation), the clusters should adopt a rod-like shape, likely concentric to the dsDNA. This is not what we have observed. The larger clusters, even if round, are likely formed by both types of binding modes, as the molecules at the edge of the cluster might be too far to bind to the dsDNA.

*We have performed a new analysis of cluster growth to gain additional insight into the mechanism. Our new data show examples of growth along the dsDNA, perpendicularly to the dsDNA (PYD-PYD binding) and combined AIM2-DNA and PYD-PYD binding. These data are included in **Supplementary Figure 10**. We have added surface plots to represent more clearly the direction of growth indicated with arrows.*

“For primary nucleation (Fig. 8b), do the authors imagine the new monomer to bind the DNA first – as is drawn and described? Would it not make more sense, in Fig. 8b, for the new monomers to first interact (possibly dynamically) via the PYD domain, which increases the local concentration, and hence next enables DNA binding, if diffusion on the DNA is indeed not occurring? Can the authors see short spikes of fluorescence that may evidence binding (PYD-PYD)? The authors describe the long clusters perpendicular to the DNA as secondary nucleation, which might make sense for a second row of AIM2 proteins as drawn in Fig. 8c, but not for such large structures. What would happen if the PYD-AIM2 proteins would be mixed with PYD monomers? Would this promote secondary nucleation, as the latter is based on PYD while primary needs both?”

*We have included new data from kymographs showing traces of a single AIM2 molecule growing to two molecules bound to dsDNA (**Figure 5**). Since the data are derived from*

kymographs where the laser scans only along the DNA, the reported growth is happening on the dsDNA molecule.

We have not observed spikes of fluorescence indicating AIM2-AIM2 binding by PYD interaction prior to dsDNA binding.

AIM2 can indeed polymerize forming filaments in the absence of dsDNA as observed by TEM¹. The filament is formed by PYD-PYD interactions leaving the HIN domains exposed¹. However, this mechanism could lead to sterile inflammation as AIM2 would form the inflammasome w/o any trigger. Thus, AIM2-AIM2 binding prior to dsDNA binding seems physiologically less relevant.

The model depicted in Figure 8 has the purpose of helping understand the two binding modes in analogy to the double nucleation mechanism, but it does not represent the details of how AIM2 polymerization happens. The PYDs of AIM2 and other Death Domain proteins, such as the inflammasome adaptor ASC, have three interacting surfaces that have been shown to be key for polymerization^{22,25}. Thus, a second row of AIM2 as depicted in Figure 8c can lead to the very large clusters observed as each PYD has two additional interacting surfaces for binding to more PYDs.

Previously published TEM data show that the PYD of AIM2, extricated from the HIN, polymerizes forming filaments¹. Our expectation is that adding the PYD to AIM2 clusters bound to dsDNA would promote oligomer growth by PYD-PYD interaction (analogous to secondary nucleation).

“Other points:”

“1) The main text gives sometimes too much detail (EG line 118-148), while not describing key things like which fluophores is used and how labelling is performed.”

The information provided in this section is necessary for the reader to understand the limitations on molecular counting based on fluorescence intensity measurements. Additional information in this section has been added in the revised version as per request of reviewer #2. However, as reviewer #3 indicates, it is very important to be more specific about the fluorophore used and how the labeling was performed. The fluorophore used is Alexa 488 and this information is indicated in the Methods Section. We have added a new sentence (line 95) in the revised manuscript in the first paragraph of the Results Section for additional clarity. The Methods Section includes “Labeling of AIM2 constructs with Alexa Fluor 488.” For the reader to find this information more readily, we have added in the main text “(Methods section)” in line 186 when mentioning the labeling efficiency.

“2) The ‘asymmetric’ on line 109 is difficult to understand / picture without more information here.”

For clarity and based on the information provided in Figure 1b of the revised manuscript, we have changed the wording of this paragraph. Please see lines 127 - 129.

“3) The concentration is clearly described in the main text and figure for 4e and f, but not for the reference 4d.”

*We apologize for this oversight. All AIM2 concentration values used to obtain the k_{off} are indicated in the legend of **Figure 3d** in the revised version, as well as in the text in lines 231 - 234 and 248 - 250.*

“4) The argumentation starting on line 257 appears quite weak to me, as the estimations of the KD values are very approximate (also given the difficulty of measuring K_{on}), and the argumentation is based on this number being closer to data of one construct compared to another, determined with a very different technique. This correspondence could be based in part on chance, and hence should not be used to draw conclusions from (line 259-263).”

We have reworded this argumentation to avoid drawing conclusions.

“5) How often were the behaviors in Figure 5 observed?”

“6) How often were the behaviors in Figure 6 observed? This question is of direct importance in judging the primary-secondary nucleation model that is central to this paper.”

Response to questions 5) and 6): *The concept of primary and secondary nucleation stems from the fact that AIM2 binds to dsDNA as a mechanism of pathogen detection and oligomerizes on the dsDNA to trigger the formation of the inflammasome, which is a large filamentous multi-protein complex. Once AIM2 binds to dsDNA (analogous to primary nucleation) polymerization takes place to form the inflammasome. The work presented in this manuscript shows for the first time that the AIM2 inflammasome can grow following two different binding modes. These two types of growth can be compared to a double nucleation mechanism in which the primary nucleation is the binding to dsDNA (for pathogen detection) and oligomerization on the DNA, and the secondary nucleation only contributes to oligomer growth. The combination of optical traps and confocal fluorescence microscopy has allowed the visualization and analysis of such events.*

*We agree with the reviewer that the frequency of growth events in directions parallel and perpendicular to the dsDNA is important to provide quantitative information on the double nucleation mechanism. Although we have not performed a statistical analysis of these events yet; growth that is purely perpendicular to the dsDNA is less abundant than growth along the dsDNA. We have included a new analysis of cluster growth in **Supplementary Figure 10**, showing representative examples of growth along the DNA, perpendicular, and a combination of the two types of growth (fluorescence intensity increasing along the dsDNA and perpendicularly). Further research is necessary to provide additional detail and quantitative information for a more in-depth understanding of inflammasome growth mechanisms.*

We have made this information clearer in lines 395 - 399 and Supplementary Figure 10.

“7) To me, the 3D images in Fig. 6d are more confusing than helpful. In particular on the left one does not see much. Maybe two overlapping ‘contour-line’ plots as used in maps is more insightful.”

*We have evaluated how to better represent these data, including the use of contour plots. We have preferred not to use the latter as they will result in numerous contours unless the threshold is sufficiently high, which could leave out new fluorescence signals indicating growth. The surface plots show the direction of growth by using two axes (parallel and perpendicular to dsDNA) as reference. To make the figures clearer, we have added arrows indicating direction of growth and highlighted the new fluorescent signals with asterisk in the surface plots shown in **Figure 6 and Supplementary Figure 10**.*

“8) Why would the growth rate be higher for AIM2-AIM2 than for AIM2-DNA growth?”

*We can speculate that AIM2-AIM2 growth only requires PYD-PYD binding, whereas HIN-DNA and PYD-PYD interactions are necessary for AIM2-DNA growth. We have included this suggestion in lines 390 - 393. The new growth analysis performed to gain additional insight on the mechanism (**Supplementary Figure 10**) indicates that AIM2-AIM2 growth is not always significantly faster than AIM2-DNA growth. Based on the new and original analyses, we have indicated in the revised version that growth rates depend on several factors such as protein concentration, starting number of molecules, oligomer shape and type of binding. This information is included in lines 341 - 360. In addition, the text on PYD-PYD binding has been modified accordingly in lines 384 - 385.*

“9) ‘permanence time’ does not seem to be an appropriate term.”

Permanence time is no longer used in the revised version and has been replaced by “residence time,” as this term is more commonly used for DNA binding proteins.

“10) The conclusion drawn in line 383-384 seems tenuous.”

The information about the basal concentration has been removed. The autoinhibitory mechanism has been previously proposed based on evidence on PYD-HIN interaction².

References

1. Morrone, S. R. *et al.* Assembly-driven activation of the AIM2 foreign-dsDNA sensor provides a polymerization template for downstream ASC. *Nat Commun* **6**, 7827 (2015).
2. Jin, T. *et al.* Structures of the HIN Domain:DNA Complexes Reveal Ligand Binding and Activation Mechanisms of the AIM2 Inflammasome and IFI16 Receptor. *Immunity* **36**, 561–571 (2012).
3. <https://pubchem.ncbi.nlm.nih.gov/compound/Alexa-488>
4. <https://pubchem.ncbi.nlm.nih.gov/compound/102227067>
5. <https://www.thermofisher.com/us/en/home/references/molecular-probes-the-handbook/tables/fluorescence-quantum-yields-and-lifetimes-for-alexa-fluor-dyes.html>
6. <https://www.sigmaaldrich.com/US/en/product/sigma/41051>
7. Loeff, L., Kerssemakers, J. W. J., Joo, C. & Dekker, C. AutoStepfinder: A fast and automated step detection method for single-molecule analysis. *Patterns* **2**, 100256 (2021).
8. Marin-Gonzalez, A., Vilhena, J. G., Perez, R. & Moreno-Herrero, F. Understanding the mechanical response of double-stranded DNA and RNA under constant stretching forces using all-atom molecular dynamics. *Proceedings of the National Academy of Sciences* **114**, 7049–7054 (2017).
9. Newton, M. D. *et al.* DNA stretching induces Cas9 off-target activity. *Nat Struct Mol Biol* **26**, 185–192 (2019).
10. Zolmajd-Haghighi, Z. & Hanley, Q. S. When One Plus One Does Not Equal Two: Fluorescence Anisotropy in Aggregates and Multiply Labeled Proteins. *Biophys J* **106**, 1457–1466 (2014).
11. Ohya, Y., Yabuki, K., Tokuyama, M. & Ouchi, T. Construction and Energy Transfer Behavior of Sequential Chromophore Arrays on an Oligo-DNA Assembly. *Supramol Chem* **15**, 45–54 (2003).
12. Runnels, L. W. & Scarlata, S. F. Theory and application of fluorescence homotransfer to melittin oligomerization. *Biophys J* **69**, 1569–1583 (1995).
13. Cuppoletti, A., Cho, Y., Park, J.-S., Strässler, C. & Kool, E. T. Oligomeric Fluorescent Labels for DNA. *Bioconjug Chem* **16**, 528–534 (2005).
14. Usman, S. & Patil, A. Radiation detector deadtime and pile up: A review of the status of science. *Nuclear Engineering and Technology* **50**, 1006–1016 (2018).
15. Cheon, N. Y., Kim, H.-S., Yeo, J.-E., Schärer, O. D. & Lee, J. Y. Single-molecule visualization reveals the damage search mechanism for the human NER protein XPC-RAD23B. *Nucleic Acids Res* **47**, 8337–8347 (2019).
16. Kong, M. *et al.* Single-Molecule Imaging Reveals that Rad4 Employs a Dynamic DNA Damage Recognition Process. *Mol Cell* **64**, 376–387 (2016).
17. Guimaraes, C. P. *et al.* Site-specific C-terminal and internal loop labeling of proteins using sortase-mediated reactions. *Nat Protoc* **8**, 1787–1799 (2013).
18. Mariusz, M., R, M. S. & Jungsan, S. Digital signaling network drives the assembly of the AIM2-ASC inflammasome. *Proceedings of the National Academy of Sciences* **115**, E1963–E1972 (2018).
19. Rhine, K., Skanchy, S. & Myong, S. Single-molecule and ensemble methods to probe RNP nucleation and condensate properties. *Methods* **197**, 74–81 (2022).

20. Hansen, J. C., Maeshima, K. & Hendzel, M. J. The solid and liquid states of chromatin. *Epigenetics Chromatin* **14**, 50 (2021).
21. Huang, Y. *et al.* Common Pitfalls and Recommendations for Using a Turbidity Assay to Study Protein Phase Separation. *Biochemistry* **60**, 2447–2456 (2021).
22. Lu, A. *et al.* Unified Polymerization Mechanism for the Assembly of ASC-Dependent Inflammasomes. *Cell* **156**, 1193–1206 (2014).
23. Ming, M. S. *et al.* Inflammasome activation causes dual recruitment of NLRC4 and NLRP3 to the same macromolecular complex. *Proceedings of the National Academy of Sciences* **111**, 7403–7408 (2014).
24. Eaton, W. A. Hemoglobin S polymerization and sickle cell disease: A retrospective on the occasion of the 70th anniversary of Pauling’s Science paper. *Am J Hematol* **95**, 205–211 (2020).
25. Oroz, J., Barrera-Vilarmau, S., Alfonso, C., Rivas, G. & de Alba, E. ASC Pyrin Domain Self-associates and Binds NLRP3 Protein Using Equivalent Binding Interfaces. *Journal of Biological Chemistry* **291**, 19487–19501 (2016).

REVIEWER COMMENTS

Reviewer #1 (Remarks to the Author):

The authors have addressed most of previous concerns from this reviewer.

In response to previous concerns, the authors generated a heat map from the kymograph in figure 5 to demonstrate the increase of fluorescence intensity with time. However, the authors selectively focus on different time windows in the three heat maps to support their conclusion that the fluorescence intensity increases, but ignore the rest of the time windows in which the intensity drops off, for example after 1 second on the top map, after 3 seconds in the middle, and after 4 seconds on the bottom. This needs to be clarified. Furthermore, why would 1 nM AIM2 concentration in the middle show faster heat map intensity increase than 10 nM AIM2 in the middle even though they both start with one AIM2 molecule bound? Lastly, it is not clear how the heat map was generated since the only information provided is "heat maps were created with a custom-made Python program". If the heat map was generated by only adding up the fluorescence intensity with time, without accounting for possible dissociation of AIM2 from DNA, then this is weak evidence to support the authors' conclusion. This needs to be clarified.

Reviewer #2 (Remarks to the Author):

The authors have made a lot of effort to address my concerns, and I appreciate that. Yet, there are still lingering issues that remain unsatisfying.

1. I do not think the authors' procedure of determining the area of clusters is appropriate. The spots produced by clusters are likely to be Gaussian in shape or close to Gaussian (certainly in the rims the intensity should fall gradually), and therefore I believe that a better procedure would involve fitting clusters to a Gaussian function and using that to determine size, rather than arbitrarily assigning pixels to the cluster. BTW, the authors overestimate what should be the sizes of objects based on the diffraction limit, there is no reason to add 2σ , certainly not to the average size of a cluster. Perhaps the distribution of sizes would have this property, but not the average size.

2. While I'm sure the authors are correct that it is possible to count their proteins when their copy numbers are small, I am less convinced about the counting when it comes to tens and even hundreds of molecules, as there is no calibration in the paper.

3. The authors surmise that the size of a single AIM2 molecule is " ~ 5 nm wide and ~ 8 nm long," but I honestly don't see how a small protein can be that large, even if it is composed of two domains.

4. It is strange that the large(r) clusters seem to be fixed in size and position. First, what happens with photobleaching in these clusters? Second, it makes little sense that these clusters are totally fixed, it is more likely that there is exchange of molecules that leave the clusters and new molecules that join, maintaining some kind of a steady state. And third, the observation that these clusters do not diffuse is interesting, but it is very puzzling that the authors seem to see no diffusing clusters at all. Multiple authors have seen diffusion on DNA, and even the paper cited by the authors shows mixed behavior, with many diffusing molecules and some static ones (but the static molecules seem to be related to specific DNA features- is there something similar here?). One would expect at least to see a transition from diffusion of small(er) clusters to no diffusion in larger ones, or a mixed behavior, but no total lack of diffusion. Is it possible that diffusing clusters are missed in the analysis? I am quite sure that ~ 3 seconds are enough to see motion on DNA. The typical diffusion coefficient of proteins on DNA can be as large as $.5-1$ micron²/sec, which would lead to significant motion in 3 seconds.

5. I believe the authors did not understand my question in point 4 of the review. I suggested that quenching is possible. In fact, it has been shown that when multiple dyes are in significant proximity, their average intensity may go down. That phenomenon, if it occurs, would lead to undercounting.

6. A minor point: the energy of red and blue photons is not relevant to the number of emitted photons. If the spectrum is shifted, and assuming the same extinction coefficient, the same number of photons would be emitted- no photons will be gained because the energy per photon is lower.

Reviewer #3 (Remarks to the Author):

I believe that the authors have properly addressed the main concerns that I had and hence I now do support publication.

This second revision includes new results, explanations, and corrections that hopefully have strengthened the quality of the manuscript and addressed the reviewers' concerns. Please, find below a point-by-point response.

Response to the reviewers' comments:

Reviewer #1:

“The authors have addressed most of previous concerns from this reviewer.

In response to previous concerns, the authors generated a heat map from the kymograph in figure 5 to demonstrate the increase of fluorescence intensity with time. However, the authors selectively focus on different time windows in the three heat maps to support their conclusion that the fluorescence intensity increases, but ignore the rest of the time windows in which the intensity drops off, for example after 1 second on the top map, after 3 seconds in the middle, and after 4 seconds on the bottom. This needs to be clarified. Furthermore, why would 1 nM AIM2 concentration in the middle show faster heat map intensity increase than 10 nM AIM2 in the middle even though they both start with one AIM2 molecule bound? Lastly, it is not clear how the heat map was generated since the only information provided is “heat maps were created with a custom-made Python program”. If the heat map was generated by only adding up the fluorescence intensity with time, without accounting for possible dissociation of AIM2 from DNA, then this is weak evidence to support the authors' conclusion. This needs to be clarified.”

Most of the regions in kymographs show intensity decreasing due to photobleaching. This information was indicated in the revised version. This behavior is also shown in Supplementary Figure 4, which illustrates a typical kymograph of an AIM2 cluster of approximately 10 molecules.

As another example of similar behavior, please see the figures below from the published work: *Real-Time Assembly of Virus like Nucleocapsids Elucidated at the Single-Particle Level*. Margherita Marchetti, Douwe Kamsma, Ernesto Cazares Vargas, Armando Hernandez García, Paul van der Schoot, Renko de Vries, Gijs J. L. Wuite, and Wouter H. Roos. *Nano Letters* 2019 19 (8), 5746-5753 DOI: 10.1021/acs.nanolett.9b02376.

Figure 3 in this work represents the accumulation of viral capsid protein on dsDNA. Several traces show an intensity decrease with time. Figure S4 (same published work) represents this behavior more clearly: an example of the intensity decay of a single trace of bound polypeptides undergoing photobleaching.

[REDACTED]

*Figures 3 and 4 retrieved from Nano Letters 2019
19 (8), 5746-5753.*

AIM2 clusters behave similarly to what is shown in the figures above. Please compare with our Supplementary Figure 4. However, we have been able to show an increase in fluorescence intensity in some regions of the kymographs.

For the top kymograph in Figure 5, dissociation does not need to be considered as we have shown that clusters of > 3 molecules do not detach. This information was indicated in the revised version. In addition, because the trace is continuous, an increase in intensity implies the association of molecules and thus cluster growth. It is clear from the original kymograph in the middle of Figure 5 that the single molecule does not detach. Thus, the overall lower intensity between the trace fragments corresponding to 1 and 2 molecules is likely due to blinking, which is a common phenomenon observed in traces. The trace at the bottom of Figure 5 (previous version) behaves similarly to the one in the middle, but the absence of intensity between fragments could indicate detachment. Therefore, this trace does not clearly show cluster growth. Because AIM2 does not bind to any specific dsDNA sequence, it would be a coincidence that a new cluster of 2 molecules binds in the same position. Nonetheless, we cannot rule out this option. For this reason, we have replaced this trace with a continuous one (second revision).

The heat maps were generated by considering the increase in fluorescence intensity. Dissociation does not need to be considered based on the number of molecules for the kymograph at the top. Because the traces are continuous (no complete detachment) an increase in intensity shows the addition of molecules (cluster growth) thus supporting the data shown in Figures 4, 6, and Supplementary Figure 10 of confocal 2D scans.

It is expected that the average cluster growth rate will increase at higher concentrations for the same number of starting molecules. We also indicated in the revised version, “In addition, the structural arrangements of the oligomers (**Figure 1b**) will dictate the different interacting possibilities, which could also affect the growth rate.” We are showing single-molecule behavior, and in the cases shown in the previous version of Figure 5, the growth is similar for the 1-molecule traces at 1 and 10 nM perhaps not matching average behavior.

We would like to emphasize we have shown many examples of cluster growth in Figures 4, 6, and new Supplementary Figure 10 in confocal 2D scans. Observing cluster growth of a polymerizing protein in kymographs is difficult mainly due to photobleaching. Despite this difficulty, we have shown actual growth in kymographs.

In the second revision, Figure 5 and its legend have been modified to clarify the aspects discussed above. The revised Methods Section includes additional information on how the heat maps were generated.

Reviewer #2:

“The authors have made a log of effort to address my concerns, and I appreciate that. Yet, there are still lingering issues that remain unsatisfying.

1. I do not think the authors’ procedure of determining the area of clusters is appropriate. The spots produced by clusters are likely to be Gaussian in shape or close to Gaussian (certainly in the rims the intensity should fall gradually), and therefore I believe that a better procedure would involve fitting clusters to a Gaussian function and using that to determine size, rather than arbitrarily assigning pixels to the cluster. BTW, the authors overestimate what should be the sizes of objects based on the diffraction limit, there is no reason to add $2 \times \sigma$, certainly not to the average size of a cluster. Perhaps the distribution of sizes would have this property, but not the average size.”

To determine cluster surface areas, we have used the selection tool from Fiji to select the pixels corresponding to the different clusters and the option to measure surface area. These surface areas would consider all pixels we observe.

Following the reviewer’s suggestion, we have fitted cluster intensities to Gaussian distributions and have obtained the radii at FWHM to determine the cluster surface areas. We have applied this method to a subset of clusters that were sufficiently separated from each other for a proper fit. The new results indicate that most cluster sizes fall in the 0.05 to 0.15 μm^2 range (new **Figure 1c**). Large clusters typically show different shapes thus Gaussian fit might not properly represent all surface areas.

We have modified the main text, Figure 1c, and the Methods section to include this new information. We have also changed the information related to the PSF by referring only to the FWHM in accordance with the new method used for cluster analysis.

2. “While I’m sure the authors are correct that it is possible to count their proteins when their copy numbers are small, I am less convinced about the counting when it comes to tens and even hundreds of molecules, as there is no calibration in the paper.”

We have determined the number of photons emitted by a single fluorophore by two different methods as explained in lines 181 – 191 to estimate the number of molecules in oligomers. We have mentioned several factors affecting molecular counting from fluorescence intensity. We have observed emission in the red, implying energy transfer processes are happening. This emission is for clusters larger than ~ 10 molecules and is constant up to ~ 52 molecules. However, self-quenching and other energy transfer processes could impact the estimated number of fluorophores in larger clusters. In this work, the most important result related to molecular counting is that clusters are typically smaller than 25 molecules with a slightly higher number of events in the 11 - 15 molecule range. We believe this result is reliable because the emission in the red is small and constant up to ~ 52 molecules, and because a similar estimation has been previously reported using different techniques. However, larger cluster sizes might be underestimated. To better inform potential readers of this possibility, we have added the sentence “The number of molecules in clusters emitting more than 575 photons is only an approximate value as energy transfer processes are happening more pronouncedly than for smaller clusters.” (Lines 198 – 200). The legend of Figure 2 in the previous version already indicated that molecular counting in larger clusters could be underestimated.

3. “The authors surmise that the size of a single AIM2 molecule is “~ 5 nm wide and ~ 8 nm long,” but I honestly don’t see how a small protein can be that large, even if it is composed of two domains.”

The dimensions of AIM2 indicated in the previous response were based on experimental and modeled structures. We explain below how these dimensions were determined.

The full-length structure (PYD and HIN domains) of AIM2 has not been reported. To estimate the length (~ 8 nm) and width (~ 5 nm) of AIM2 that was reported in our previous response, we created a molecular model using the program I-TASSER (<https://zhanggroup.org/I-TASSER/>): a protein structure prediction program that uses PDB structures as templates. In addition, we analyzed the structures of the individual domains that are reported in the PDB: AIM2-HIN bound to DNA (PDB 3RN2) and AIM2-PYD tagged to MBP (PDB 3VD8).

We show below actual screenshots of inter-atom distance calculations by the software Chimera X (<https://www.cgl.ucsf.edu/chimerax/>) of the three structures: the model of full-length AIM2, and the PDB structures of the HIN, and PYD domains.

The red dashed line in all figures connects two selected atoms to provide the dimensions. As it is not very clear from the screenshots, we have highlighted the red line with red arrows. The inter-

Editorial Note: With reference to screenshots on p40-41 of this Peer Review File, molecular graphics and analyses performed with UCSF ChimeraX, developed by the Resource for Biocomputing, Visualization, and Informatics at the University of California, San Francisco, with support from National Institutes of Health R01-GM129325 and the Office of Cyber Infrastructure and Computational Biology, National Institute of Allergy and Infectious Diseases.

UCSF ChimeraX: Tools for structure building and analysis. Meng EC, Goddard TD, Pettersen EF, Couch GS, Pearson ZJ, Morris JH, Ferrin TE. *Protein Sci.* 2023 Sep 29:e4792. Online ahead of print.

Page | 5

atom distance calculated by Chimera X is shown at the right of the screenshot pointed by a red arrow.

The inter-atomic distance between the two selected atoms in the PYD and HIN of AIM2 from the molecular model is close to 8.5 nm:

The inter-atomic distance between two atoms at the edges of the AIM2-HIN domain from the reported PDB 3RN2 is 5.1 nm:

The inter-atomic distance between two atoms at the edges of the AIM 2 PYD tagged to MBP from the reported PDB 3VD8 is ~ 4 nm:

4. It is strange that the large(r) clusters seem to be fixed in size and position. First, what happens with photobleaching in these clusters? Second, it makes little sense that these clusters are totally fixed, it is more likely that there is exchange of molecules that leave the clusters and new molecules that join, mainlining some kind of a steady state. And third, the observation that these clusters do not diffuse is interesting, but it is very puzzling that the authors seem to see no diffusing clusters at all. Multiple authors have seen diffusion on DNA, and even the paper cited by the authors shows mixed behavior, with many diffusing molecules and some static ones (but the static molecules seem to be related to specific DNA features- is there something similar here?). One would expect at least to see a transition from diffusion of small(er) clusters to no diffusion in larger ones, or a mixed behavior, but no total lack of diffusion. Is it possible that diffusing clusters are missed in the analysis? I am quite sure that ~3 seconds are enough to see motion on DNA. The typical diffusion coefficient of proteins on DNA can be as large as .5-1 micron²/sec, which would lead to significant motion in 3 seconds.”

Lines 170 and 175 of the manuscript explain why photobleaching effects in 2D scans can be considered negligible. Photobleaching is more pronounced in movies due to continuous acquisition. A photoprotection cocktail has been always present in all experiments.

We would like to refer to our answer to questions #1 and #2 from reviewer #3 in our previous response to answer this new question from review #2 on monomer exchange in and out of the clusters. We included results from new experiments and figures from published papers in our answer.

In our manuscript, we compare our observations on AIM2 with results reported for IFI16 (a protein sensor with two HIN domains and a PYD). The latter can diffuse several micrometers along the dsDNA when clusters are smaller than ~ 8 molecules. Larger clusters are static.

Diffusing AIM2 clusters were not missed. We have analyzed hundreds of AIM2 single-molecule traces and hundreds of traces corresponding to AIM2 oligomers of different sizes bound to multiple positions on the dsDNA. We have not observed in AIM2 the behavior reported for IFI16. These results do not seem to be related to any position on the dsDNA. We need to continue studying why AIM2 is not capable of similar behavior. We hypothesize that specific structural characteristics of the HIN domains play an important role. Perhaps interdomain dynamics in IFI16 with 2 HINs have an effect. AIM2 only has one HIN and thus interdomain dynamics will not be a contributing factor.

We have included lines 494 to 496 (below) in the second revision to complement the previous explanations and to point out that this issue requires further investigation.

“Interdomain dynamics of IFI16 HIN domains, which are absent in AIM2, could play a role in diffusion. Additional studies are required to fully understand the different diffusion behavior of dsDNA sensors.”

[REDACTED]

[REDACTED]

5. I believe the authors did not understand my question in point 4 of the review. I suggested that quenching is possible. In fact, it has been shown that when multiple dyes are in significant proximity, their average intensity may go down. That phenomenon, if it occurs, would lead to undercounting.

In the previous revised version of the manuscript, we stated the following:

“Evidence show that fluorophore clustering affects the emission properties of individual fluorophores due to energy transfer processes between them³²⁻³⁴. In fact, homo-FRET³⁵ and Stokes shift processes³⁶ resulting from fluorophore clustering have been observed. Therefore, **the total emitted light of fluorophores clustered in proximity is frequently enhanced or decreased relative to the expected intensity from the total number of fluorophores**^{32,37}. For these reasons, fluorophore clustering poses challenges in determining the stoichiometry of protein complexes based on fluorescence intensity³⁷⁻⁴⁰.”

When the fluorescence intensity decreases due to fluorophore oligomerization, we can consider this process as self-quenching, but the intensity does not always decrease. Please, see the abstract below from the published work; Cuppoletti, A., Cho, Y., Park, J.-S., Strässler, C. & Kool, E. T. Oligomeric Fluorescent Labels for DNA. *Bioconjug Chem* **16**, 528–534 (2005).

“In an effort to find fluorescent labels that have large Stokes shifts and increased emission intensity, a strategy for fluorescence labeling of DNA was explored in which multiple individual fluorophores are incorporated at adjacent positions at the end of a DNA probe. To encourage close interactions, hydrocarbon and heterocycle fluorophores were substituted at C-1 of deoxyribose, replacing the DNA base. The C-glycosides studied contained the well-known fluorophores terphenyl, pyrene, and terthiophene. For comparison, a commercial fluorescein-dU nucleotide was examined. Oligomeric labels containing up to five fluorophores were tested. Interestingly, all four dyes behaved differently on multiple substitution. **Fluorescein displayed strong self-quenching properties**, with the quantum yield dropping severalfold with each additional substitution and with a constant, small Stokes shift. In contrast, **pyrene showed increases in quantum yield on**

addition of more than one fluorophore and yielded efficient long-wavelength emission on multiple substitution, with Stokes shifts of >130 nm. Oligomeric terphenyl labels gave a small progressive red shift in absorption and a marked red shift in emission wavelength and showed a strong increase in brightness with more monomers. Finally, terthiophene oligomers showed self-quenching combined with increasing Stokes shifts. Overall, the results suggest that some oligomeric fluorescent labels exhibit properties not available in common fluorescein class (or other commercial) labels, such as large Stokes shifts and increasing brightness with increasing substitution.”

This work is cited in the manuscript.

Please see our answer to point #2 related to this effect on clusters emitting more than 575 photons.

6. A minor point: the energy of red and blue photons is not relevant to the number of emitted photons. If the spectrum is shifted, and assuming the same extinction coefficient, the same number of photons would be emitted- no photons will be gained because the energy per photon is lower.

Yes. Thank you for pointing this out.

We know that 11 - 12 blue photons correspond to 1 fluorophore based on our data. We will assume that the average of 14 red photons corresponds to 1 fluorophore.

We have modified the text, accordingly, in lines 164 – 168 and 189 – 191.

REVIEWERS' COMMENTS

Reviewer #1 (Remarks to the Author):

The authors have sufficiently addressed this reviewer's concerns and the manuscript may be suitable for publication.

Reviewer #2 (Remarks to the Author):

The authors have answered my concerns quite satisfactorily. I do not agree with every statement they make, but it is the nature of the scientific process that there is no complete agreement and some issues continue to be discussed and argued.

I therefore recommend publication of the paper.